# Disordered regions in the IRE1α ER lumenal domain mediate its stress-induced clustering

Paulina Kettel [1,2,9], Laura Marosits [1,3,9], Elena Spinetti [4,5], Michael Rechberger [1], Caterina Giannini [6], Philipp Radler[6], Isabell Niedermoser [1,3], Irmgard Fischer[1], Gijs A Versteeg [1,7], Martin Loose[6], Roberto Covino[4,8] & G Elif Karagöz [1,3] ✉

## Abstract

**Conserved signaling cascades monitor protein-folding homeostasis to ensure proper cellular function. One of the evolutionary conserved key players is IRE1, which maintains endoplasmic reticulum (ER) homeostasis through the unfolded protein response (UPR). Upon accumulation of misfolded proteins in the ER, IRE1 forms clusters on the ER membrane to initiate UPR signaling. What regulates IRE1 cluster formation is not fully understood. Here, we show that the ER lumenal domain (LD) of human IRE1α forms biomolecular condensates in vitro. IRE1α LD condensates were stabilized both by binding to unfolded polypeptides as well as by tethering to model membranes, suggesting their role in assembling IRE1α into signaling-competent stable clusters. Molecular dynamics simulations indicated that weak multivalent interactions drive IRE1α LD clustering. Mutagenesis experiments identified disordered regions in IRE1α LD to control its clustering in vitro and in cells. Importantly, dysregulated clustering of IRE1α mutants led to defects in IRE1α signaling. Our results revealed that disordered regions in IRE1α LD control its clustering and suggest their role as a common strategy in regulating protein assembly on membranes.**

**Keywords** Unfolded Protein Response; IRE1; Supported Lipid Bilayers; Biomolecular Condensates
**Subject Categories** Organelles; Signal Transduction; Translation & Protein Quality

## Introduction

The endoplasmic reticulum (ER) controls various fundamental cellular functions ranging from folding and quality control of secreted and membrane proteins to lipid biogenesis. A set of conserved signaling pathways, collectively known as the unfolded protein response (UPR), maintains ER homeostasis (Karagoz et al,

2019). IRE1, which is a single-pass ER transmembrane kinase/RNase, drives the most conserved UPR pathway (Cox et al, 1993; Shamu et al, 1994; Cox and Walter, 1996; Sidrauski et al, 1996; Cox et al, 1997). In response to ER stress, IRE1 assembles into clusters, which brings its cytosolic kinase and RNase domains in close proximity allowing for trans-autophosphorylation of the kinase domains and subsequent allosteric activation of its RNase domain (Credle et al, 2005; Aragon et al, 2009; Korennykh et al, 2009; van Anken et al, 2014). IRE1's RNase activity initiates the nonconventional splicing of the mRNA encoding the transcription factor XBP1. The spliced form of *XBP1* mRNA drives expression of the genes involved in restoring ER homeostasis, including chaperones (Cox et al, 1993; Cox and Walter, 1996; Sidrauski et al, 1996; Yoshida et al, 1998, Yoshida et al, 2001; Lee et al, 2003; Acosta-Alvear et al, 2007; Yamamoto et al, 2007; Korennykh et al, 2009). In metazoans, IRE1 activation also leads to the degradation of ER-bound mRNAs in a process known as regulated IRE1-dependent mRNA decay (RIDD), which decreases the ER protein-folding burden to alleviate ER stress (Hollien and Weissman, 2006; Hollien et al, 2009). Dysregulation of IRE1α signalling contributes to progression of various diseases including cancer and metabolic diseases (Buchan et al, 2013; Cubillos-Ruiz et al, 2017; Song et al, 2018; Harnoss et al, 2019; Harnoss et al, 2020; Lemmer et al, 2021). Therefore, understanding the mechanistic principles of IRE1α activation is of crucial importance.

IRE1 senses various perturbations to ER homeostasis to initiate the UPR, but the molecular mechanism of how this is achieved is only partially understood. Under steady-state conditions, the ER chaperone BiP was suggested to keep IRE1 in an inactive state (Bertolotti et al, 2000). Accumulation of misfolded proteins in the ER result in the dissociation of BiP from IRE1 (Bertolotti et al, 2000; Zhou et al, 2006; Oikawa et al, 2009; Amin-Wetzel et al, 2017). Under those conditions, misfolded proteins accumulating in the ER bind IRE1's lumenal domain (LD) as ligands and trigger its oligomerization (Gardner and Walter, 2011; Karagoz et al, 2017; Sundaram et al, 2018). IRE1 is also activated by lipid bilayer stress through its transmembrane domain (Volmer et al, 2013; Halbleib et al, 2017; Kono et al, 2017). As IRE1 activation correlates with its assembly into microscopically visible clusters in cells (Aragon et al,

[1]Max Perutz Laboratories Vienna, Vienna BioCenter, Vienna, Austria. [2]Vienna BioCenter PhD Program, Doctoral School of the University of Vienna and Medical University of Vienna, Vienna, Austria. [3]Medical University of Vienna, Vienna, Austria. [4]Frankfurt Institute for Advanced Studies, Frankfurt, Germany. [5]Institute of Biophysics, Goethe University, Frankfurt, Germany. [6]Institute of Science and Technology Austria, Klosterneuburg, Austria. [7]Department of Microbiology, Immunobiology and Genetics, University of Vienna, Vienna, Austria. [8]IMPRS on Cellular Biophysics, Frankfurt, Germany. [9]These authors contributed equally: Paulina Kettel, Laura Marosits.
✉E-mail: guelsuen.karagoez@meduniwien.ac.at

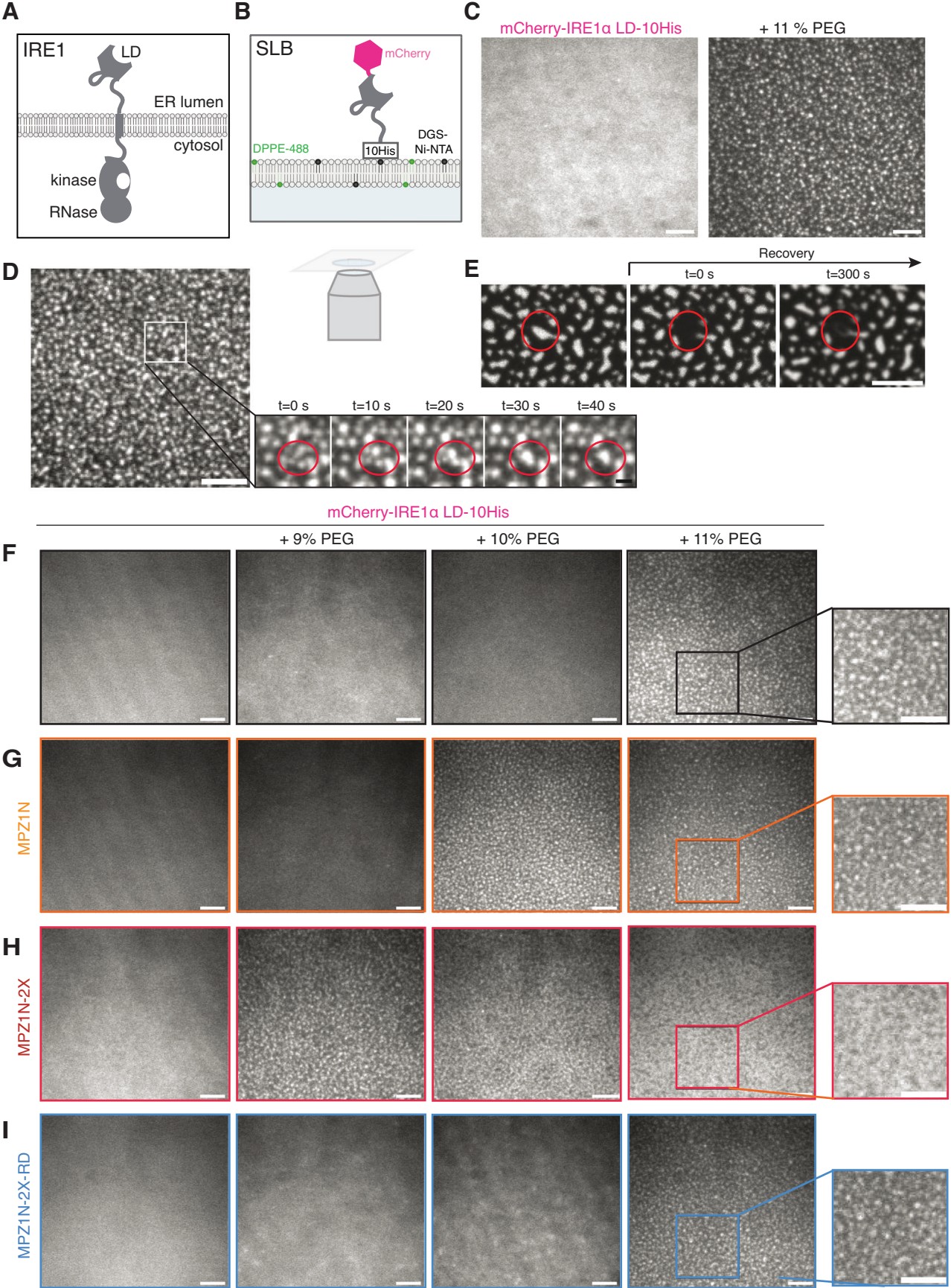

◄ **Figure 1. IRE1α LD forms clusters on supported lipid bilayers (SLB).**

(A) Schematic illustration of IRE1α domain architecture within the ER membrane. (B) Schematic illustration of the SLB setup. (C) TIRF images of mCherry-IRE1α LD-10His clustering on an SLB in the absence (left) and presence of 11% PEG. Scale bar = 5 μm. (D) Fusion events of mCherry-IRE1α LD-10His clusters on SLBs at the indicated time points. Scale bar = 5 μm, zoom in scale bar = 1 μm. Red circles display fusion of clusters. (E) FRAP images of mCherry-IRE1α LD-10His on SLBs in presence of 11% PEG within 300 s. Scale bar = 5 μm. Red circles indicate the photobleached cluster. (F) TIRF images displaying clustering of mCherry-IRE1α LD-10His tethered to SLBs via 1% Ni-NTA lipids in the presence of the indicated concentrations of PEG. Clustering is visible by the formation of fluorescent intense spots. Scale bar = 5 μm. (G) TIRF images displaying clustering of mCherry-IRE1α LD-10His in the presence of PEG and 10 μM model unfolded polypeptide ligand MPZ1N. Scale bar = 5 μm. (H) TIRF images displaying the phase diagram of mCherry-IRE1α LD-10His in the presence of PEG and 1 μM model unfolded polypeptide ligand MPZ1N-2X. Scale bar = 5 μm. (I) TIRF images displaying the phase diagram of mCherry-IRE1α LD-10His in the presence of the indicated concentrations of PEG and 1 μM control peptide MPZ1N-2X-RD. Scale bar = 5 μm.

2009; Li et al, 2010; Karagoz et al, 2017; Ricci et al, 2019; Belyy et al, 2020), its oligomerization is thought to be an important step in the activation of the pathway.

Some studies have suggested that oligomerization of IRE1 is initiated by its ER-lumenal sensor domain (Credle et al, 2005; van Anken et al, 2014; Karagoz et al, 2017; Ricci et al, 2019; Belyy et al, 2022) and mutations introduced to the oligomerization interface in IRE1's LD impair the formation of high-order oligomers and abolish IRE1 signaling in cells (Credle et al, 2005; Karagoz et al, 2017; Ricci et al, 2019; Belyy et al, 2022). Oligomerization is a conserved property of IRE1 LD from yeast to humans (Credle et al, 2005; Karagoz et al, 2017). However, previous in vitro experiments have failed to uncover the molecular basis of IRE1α LD oligomerization. While the core folded domain of human IRE1α LD (cLD) forms discrete dimers, which in a concentration-dependent manner assemble into dynamic high-order oligomers (Karagoz et al, 2017), the crystal structure of human IRE1α cLD did not display functional oligomerization interfaces (Zhou et al, 2006). Accordingly, the structural basis for IRE1α LD oligomerization has remained elusive (Zhou et al, 2006). Mutational analyses based on crosslinking coupled to mass spectroscopy data identified a hydrophobic segment in IRE1α cLD that controls its oligomerization in vitro and its clustering in cells (Karagoz et al, 2017). Yet, this method could not map the interfaces contributing to the formation of high-order oligomers. To better understand how IRE1α assembles into signaling competent clusters, more comprehensive in vitro reconstitution experiments in combination with structural studies are essential. Surprisingly, even though IRE1α is a transmembrane protein, how its physiological orientation on the membrane impacts its clustering remains unexplored.

Here, to mechanistically dissect how IRE1α LD assembles into high-order oligomers, we reconstituted IRE1α LD clustering in solution and on supported lipid bilayers (SLB) as model membranes. We revealed that disordered regions in IRE1α LD control its assembly into dynamic biomolecular condensates. Our data suggest that membranes and unfolded polypeptide ligands act synergistically in stabilizing dynamic IRE1α LD condensates into long-lived clusters to transmit the signal across the ER membrane. We propose that assembly through dynamic disordered regions might present a common strategy for protein clustering on membranes.

## Results

### IRE1α LD forms stable clusters on synthetic membranes

To investigate whether membrane association influences IRE1α LD clustering, we reconstituted the system in vitro using purified

human IRE1α LD tethered to supported lipid bilayers (SLBs). SLBs are constituted of planar membranes formed on solid surfaces which are widely used as membrane-mimics (Fig. 1A,B).

We reconstituted SLBs composed primarily of 1-palmitoyl-2-oleoyl-glycero-3-phosphocholine (98.92 mol% POPC). We used 1 mol% nickel–nitrilotriacetic acid (Ni-NTA) lipids to tether mCherry-IRE1α LD-10His to SLBs through its C-terminal 10His tag, which allows placing the entire LD of IRE1α (aa 24-443) in the topologically correct orientation (Fig. 1B). To monitor SLB integrity and fluidity, we used 0.08 mol% Atto488 labeled 1,2-Dipalmitoyl-sn-glycero-3-phosphoethanolamine (DPPE). Fluorescence recovery after photobleaching (FRAP) experiments of mCherry-IRE1α LD-10His demonstrated that mCherry-IRE1α LD-10His diffused on the SLB surface (Figs. 1C, left and EV1A; Appendix Table S1). Similarly, FRAP of Atto488 labeled DPPE lipids confirmed that the membrane was fluid (Fig. EV1A; Appendix Table S1).

To mimic the crowding of the ER environment in our in vitro assays, we used the molecular-crowding agent polyethylene glycol 8000 (PEG) (Kuznetsova et al, 2014). We monitored the behavior of mCherry-IRE1α LD-10His on the membrane surface via total internal reflection fluorescence (TIRF) microscopy and FRAP experiments at various PEG concentrations (Figs. 1C, right, F and EV1B,C). We calculated the diffusion coefficient from fluorescence recovery half-life times as previously described (Axelrod et al, 1976) (Soumpasis, 1983). Interestingly, increasing the PEG concentration in solution gradually decreased the mobile fraction and diffusion rates of membrane-bound mCherry-IRE1α LD-10His from 0.18 μm²/s without PEG to 0.02 μm²/s in presence of 11% (w/v) PEG (Fig. EV1A,B). In the presence of 10% PEG, mCherry-IRE1α LD-10His displayed a diffuse fluorescence signal (Fig. 1F), while 11% PEG induced the formation of large mCherry-IRE1α LD-10His clusters on the SLB (Fig. 1C, right, F; Movie EV1; Appendix Table S2). In the presence of 11% PEG, both mCherry-10His control and Atto488-labeled DPPE retained their dynamic behavior confirming that the integrity of the SLB was not compromised and clustering is specific to IRE1α LD (Fig. EV1C–F; Appendix Table S2). Under those conditions, mCherry-IRE1α LD-10His clusters formed and fused over time (Fig. 1D; Movie EV2). Yet, FRAP experiments showed that photobleached IRE1α LD clusters did not recover even after 300 s (Figs. 1E and EV1B,C). Instead, we observed a slight increase in mCherry-IRE1α LD-10His fluorescence at the periphery of the clusters (Fig. 1E) indicating that membrane-tethered IRE1α LD assembles into stable clusters driven by molecular crowding. To test whether membrane-tethered IRE1α LD clusters are not just aggregates, we performed wash-out experiments in which we removed the crowding agent from the

well. Removal of PEG led to the disappearance of IRE1α LD clusters back to a diffuse fluorescence signal (Fig. EV1G). Importantly, clusters could reform by adding 11% PEG, indicating that they are dynamic and reversible. Altogether, we found that IRE1α LD forms stable but reversible clusters on synthetic membranes. Notably, our data are in agreement with the FRAP experiments performed with IRE1α in cells indicating that IRE1α LD reconstituted on membranes recapitulates the physical properties of IRE1α assemblies in cells (Belyy et al, 2020).

## Binding of unfolded model polypeptides enhances IRE1α LD clustering

IRE1α's LD binds unfolded peptides that are enriched in arginine, aromatic and hydrophobic residues as a means of recognizing aberrant protein conformations (Gardner and Walter, 2011; Karagoz et al, 2017). We next tested whether binding of model unfolded polypeptides would enhance IRE1α LD clustering on SLBs. We used peptides that we had previously shown to interact with IRE1α LD (Karagoz et al, 2017). The binding peptides with the highest affinity were derived from Myelin Protein Zero (MPZ) referred to as MPZ derivatives. MPZ1N is a 12mer peptide with a single binding site for IRE1α LD and binds IRE1α LD with an approximate affinity of 20 μM (Fig. EV1H,I). MPZ1N-2X consists of two MPZ1N 12mers arranged in tandem, and it binds IRE1α LD with 2 μM affinity due to avidity (Karagoz et al, 2017) (Fig. EV1I). As a control, we mutated arginine residues in MPZ1N-2X to impair its interaction with IRE1α LD, yielding MPZ1N-2X-RD (Karagoz et al, 2017). Using fluorescence anisotropy experiments, we confirmed the MPZ1N-2X-RD interaction with IRE1α LD is largely impaired (Fig. EV1I).

In the stressed ER, IRE1α LD clustering may be initiated by specific interactions of IRE1α LD with un/misfolded proteins, and a bulk increase in molecular crowding due to blocked secretion of un/misfolded proteins. Therefore, we next tested whether IRE1α LD's interactions with model unfolded polypeptides would decrease the threshold for its clustering in the presence of a crowding agent. We found that incubation with peptides reduced the effective concentration of PEG required to drive the clustering of mCherry-IRE1α LD-10His (Figs. 1F–H and EV1J). This increased propensity was specific, as incubation of mCherry-IRE1α LD-10His with the mutant peptide MPZ1N-2X-RD did not impact its clustering (Fig. 1I). Importantly, FRAP experiments revealed that the peptides did not impair SLB integrity (Figs. 1F–I and EV1K,L; Appendix Fig. 1 and Appendix Table S2). In sum, we succeeded in reconstituting ligand-enhanced IRE1α LD clustering on synthetic membranes from minimal components, thus recapitulating a critical step of the UPR.

## IRE1α LD forms dynamic condensates in solution

Our data suggested that IRE1α LD tethered to synthetic membranes forms stable clusters due to restricted conformational freedom on planar surfaces. This model predicts that IRE1α LD clusters formed in solution should exhibit a more dynamic behavior when compared to those formed on SLBs. To test this prediction, we monitored IRE1α LD clustering in solution by differential interference contrast (DIC) microscopy. The systematic analyses of protein concentrations and buffer conditions by DIC showed

that in the presence of 6% PEG, 12.5 μM IRE1α LD formed droplets in solution. IRE1α LD droplets resembled biomolecular condensates formed through liquid-liquid phase separation (LLPS). Both the number and size of the condensates increased at higher protein concentrations (Figs. 2A and EV2A,B). IRE1α LD formed condensates in the presence of other crowding agents (17% Ficoll400) suggesting that molecular crowding induces LLPS of IRE1α LD (Fig. EV2C). IRE1α LD condensates displayed dynamic and liquid-like behavior in solution, as evidenced by fusion events (Fig. 2A,B; Movie EV3). FRAP experiments confirmed the liquid-like nature of IRE1α LD condensates and revealed that IRE1α LD molecules exchanged in and out of the condensates with a mobile fraction of 82% ($t_{1/2} = 169$ s); Fig. 2C; Movie EV2). It has been observed that if a protein goes through LLPS, the liquid droplets will wet the glass surface, whereas hydrogels or less dynamic condensates do not wet solid surfaces or change shape (Wang et al, 2019). IRE1α LD condensates wetted the bottom of the glass surface in a time-dependent manner, confirming their liquid-like properties (Fig. EV2D). Neither IRE1α LD^{D123P} mutant, which is impaired in dimerization (Zhou et al, 2006), nor the mCherry control formed condensates (Fig. EV2E). This suggested that D123 is required for the formation of larger IRE1α LD clusters in solution. Altogether, our data revealed that in solution IRE1α LD forms dynamic condensates upon molecular crowding. These data suggested that tethering IRE1α LD to membranes leads to the stabilization of IRE1α LD assemblies. It is plausible that the restriction of IRE1α LD's degree of freedom, or membrane-induced structural rearrangements stabilize interfaces, which are crucial for its clustering. Consequently, this could drive the formation of long-lived IRE1α LD assemblies on membranes.

We next characterized the impact of unfolded polypeptides on the formation and dynamics of IRE1α LD condensates. Fluorescein labeled MPZ1N-2X efficiently partitioned into preformed IRE1α LD condensates, revealing that they recruit client proteins (Fig. EV2F). Instead, the Fluorescein-MPZ1N-2X-RD control peptide was not enriched in the condensates (Fig. EV2G). In an experimental condition where IRE1α LD barely formed condensates (Fig. 2D, left panel), its incubation with stoichiometric amounts of model unfolded peptides led to the formation of large condensates (Figs. 2D and EV2H). Instead, the control peptide MPZ1N-2X-RD did not impact IRE1α LD phase separation (Figs. 2D, right panel and EV2H). Incubation with MPZ1N did not lead to LLPS of IRE1α LD^{D123P} indicating that dimer formation is necessary for peptide induced LLPS (Fig. EV2I). Importantly, model unfolded polypeptides did not undergo phase separation under those conditions (Fig. EV2J,K). This data indicated that specific interactions with unfolded polypeptides facilitate IRE1α LD phase separation. We next assessed whether unfolded polypeptide-binding would impact the dynamics of IRE1α LD assemblies within condensates. FRAP experiments showed that, while MPZ1N did not significantly affect the half-time recovery of IRE1α LD after photobleaching, binding of the MPZ1N-2X peptide led to an increase in the recovery time of IRE1α LD (Figs. 2C and EV2L,M; Appendix Table S3). These data revealed that a peptide with a single binding site shifts IRE1α LD to a conformation that favors its clustering consistent with previous findings (Karagoz et al, 2017). MPZ1N-2X, which has two binding sites, can nucleate clusters by bridging IRE1α LD molecules and further stabilize IRE1α LD assemblies. We speculate that binding of unfolded polypeptides

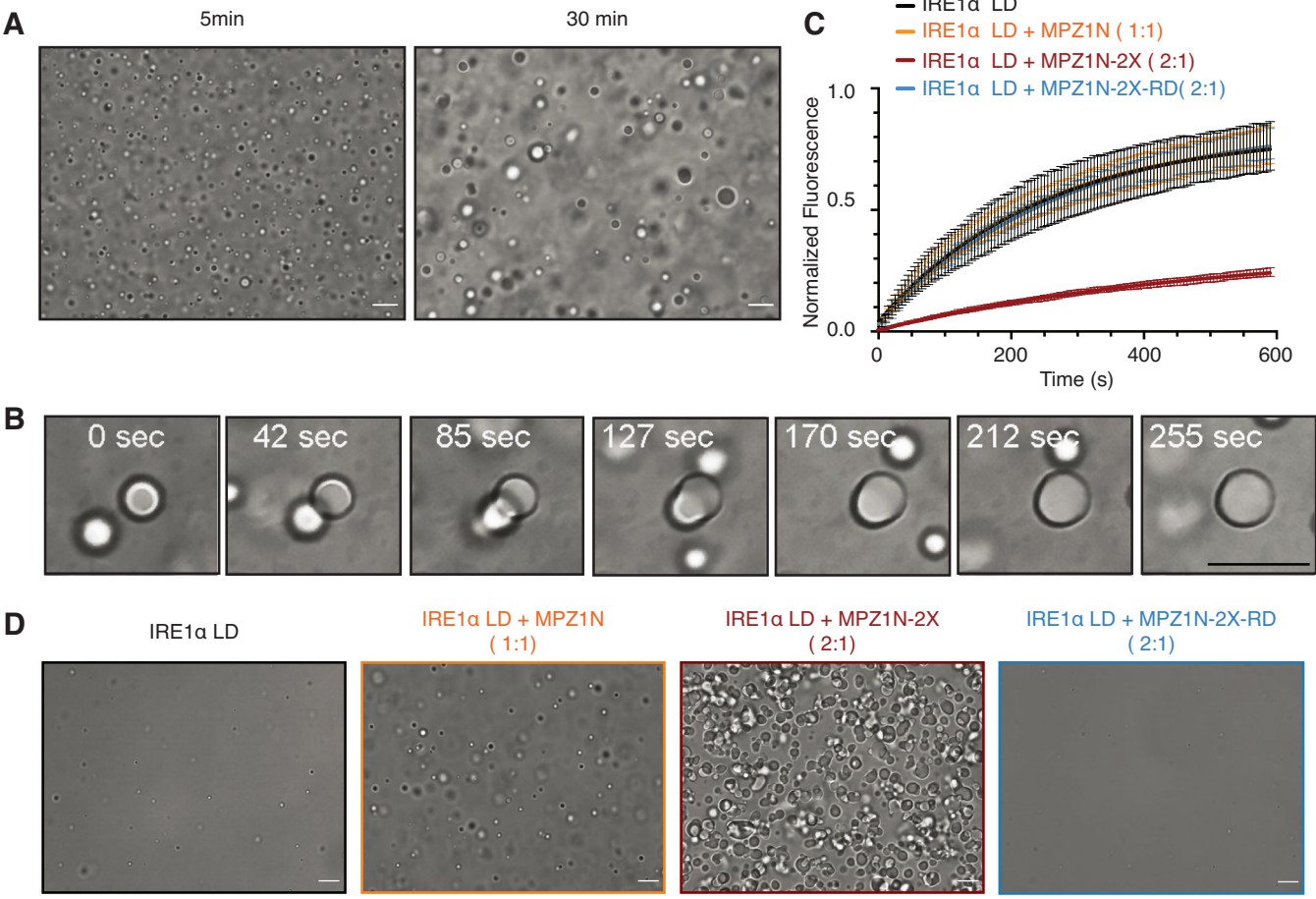

**Figure 2. IRE1α LD forms dynamic condensates in solution.**

(A) DIC microscopy images showing IRE1α LD condensates imaged after 5 min (left) and 30 min incubation with PEG (50 μM IRE1 LD, 6% PEG). Scale bar for all images = 10 μm. (B) Fusion of IRE1α LD condensates imaged by DIC microscopy. The condensates were imaged after 30 min incubation with PEG at the indicated time points (50 μM IRE1 LD, 6% PEG). Scale bar = 10 μm. (C) FRAP curves showing normalized fluorescent recovery of IRE1α LD condensates after 30 min incubation with 6% PEG. IRE1α LD (25 μM IRE1 LD, 6% PEG) in the absence (black curve) and in the presence of MPZ1N peptide (1:1 stoichiometry, orange curve), in the presence of MPZ1N-2X peptide (2:1 stoichiometry, dark red curve), in the presence of MPZ1N-2X-RD peptide (2:1 stoichiometry, blue curve). Curve marks show the mean value, error bars display the standard deviation. n = 3 independent experiments were performed while 3 condensates were bleached each experiment. (D) DIC microscopy images displaying LLPS behavior of IRE1α LD alone (50 μM IRE1α LD, 5% PEG) (left) and IRE1α LD in complex with MPZ1N (1:1 stoichiometry), MPZ1N-2X (2:1 stoichiometry) and the control MPZ1N-2X-RD. Images were taken 30 min after induction of phase separation with PEG. Scale bar = 10 μm.

with various stoichiometry and biochemical properties could modulate the dynamics of IRE1α assemblies, ultimately influencing UPR signaling in cells.

## Disordered regions in IRE1α LD drive dynamic clustering

The generation of biomolecular condensates by IRE1α LD in solution prompted us to ask which molecular interactions might explain this behavior. The formation of biomolecular condensates is often controlled by disordered regions in proteins (Hyman and Brangwynne, 2011). IRE1α LD comprises a mostly folded N-terminal motif (aa 24–307) joined to the transmembrane helix by a disordered region (aa 307–443) (Fig. 3A; Appendix Fig. 2A,B) (Erdos and Dosztanyi, 2020). In the crystal structure of IRE1α cLD (aa 24–390, pdb: 2hz6 (Zhou et al, 2006)), several segments (i.e. aa) 131–152, 307–358, and 369–390) are not resolved due to their flexibility (Fig. 3B; Appendix Fig. 2A,B). We refer to the

disordered regions in IRE1α LD as Disordered Region 1 (DR1, aa 131–152), Disordered Region 2 (DR2, aa 307–358), Disordered Region 3, (DR3, aa 369–389), and the linker region (aa 390–443), respectively. Here, we employed molecular dynamics (MD) simulations to characterize their conformation and interaction.

Atomistic MD simulations of the IRE1α cLD dimer (residues aa 29–368) revealed that DR1 and DR2 remain highly disordered during a 1 μs long simulation, not adopting any distinct secondary structures (Fig. 3C). DR2 was the most flexible part of the dimer. These data are in line with published hydrogen-deuterium exchange experiments (Amin-Wetzel et al, 2019). We then performed coarse-grained MD simulations to test whether the disordered regions might self-associate (Fig. 3D,E). We observed that DR1 did not form clusters in a 20 μs-long simulation (Appendix Fig. 2C). Instead, DR2 and the linker region readily clustered after 1 μs of simulation. The clusters were highly dynamic,

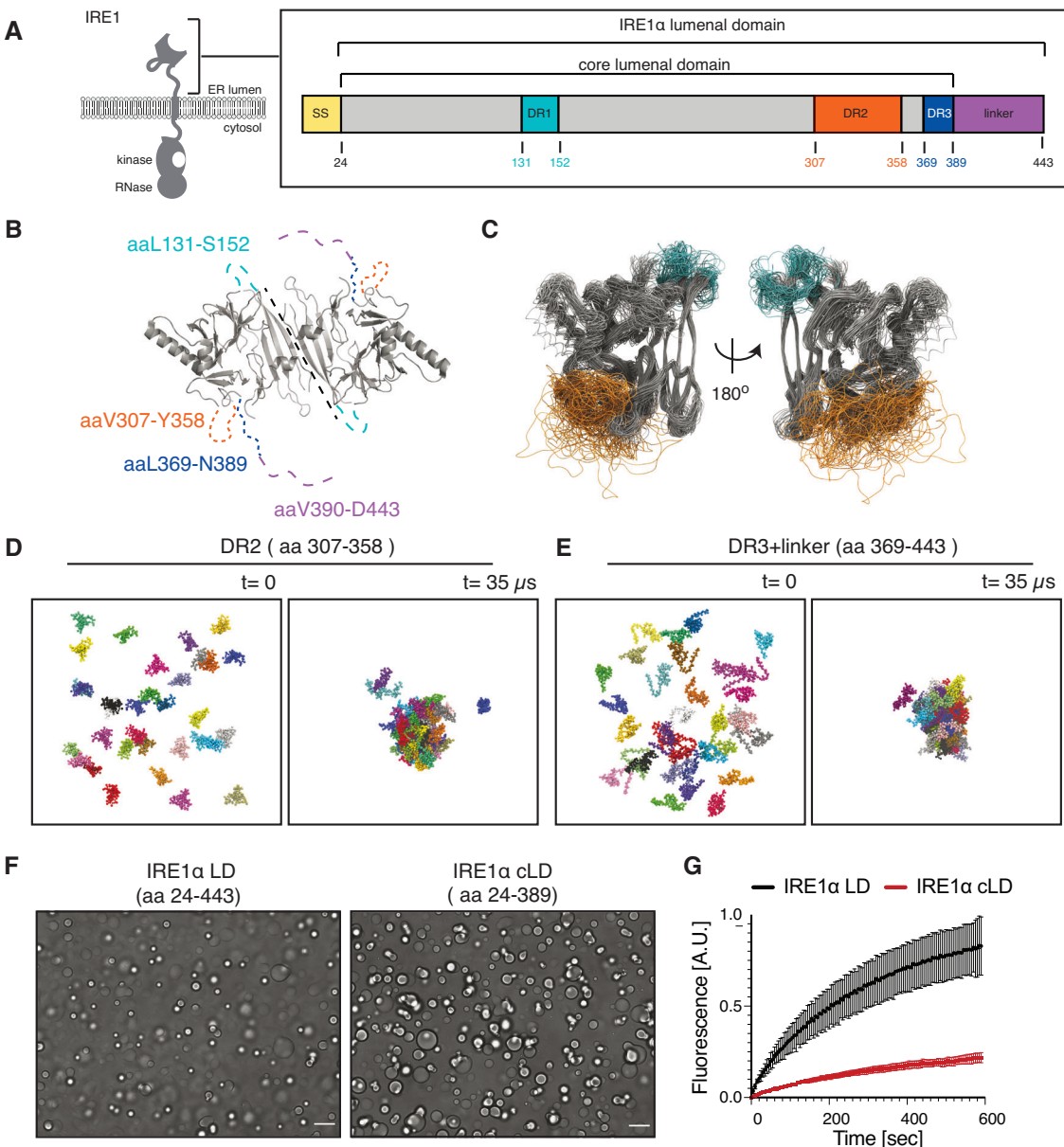

**Figure 3. Disordered regions in IRE1α LD have potential to form clusters.**

(A) Schematic description of DRs in IRE1α LD and boundaries of the core LD. The numbers correspond to the amino acid number at the domain boundaries. SS Signal sequence, DR disordered region. (B) IRE1α cLD dimeric structure based on the crystal structure of human IRE1α (pbd: 2hz6). The DRs that are not resolved in the structure are depicted by dashed lines. (C) Superposition of frames of an all-atom cLD simulation. Molecular dynamic simulations of IRE1α cLD shows flexibility of the DRs at 600 ns time scale. (D) Molecular Dynamics Simulations of 33 copies of DR2. These simulations reveal that DR2 forms clusters. (E) Molecular Dynamics Simulations of 33 copies of the linker region. These simulations reveal that the linker region forms clusters. (F) DIC images of IRE1α LD (left) and IRE1α cLD (right) reveal that IRE1α cLD is sufficient to form condensates. All images were obtained for 50 μM protein after incubation with 6% PEG for 30 min. Scale bar = 10 μm. (G) FRAP curve showing the time-dependent, normalized fluorescent recovery of 25 μM IRE1α LD (black) and IRE1α cLD (red) condensates after 30 min incubation with 6% PEG. Curve marks show the mean value, error bars display the standard deviation and the values are fitted to a one-phase association curve displaying a lower mobile fraction and longer half-life time for the IRE1α cLD condensates. *n* = 3 independent experiments were performed where 3 condensates were bleached each experiment.

and we observed reversible association of single polypeptide chains. To test the potential of heterologous associations, we simulated a system containing DR1 and DR2 and another system comprising DR2 and the linker region (Appendix Fig. 2C,D). DR2 clusters did not interact with DR1 segments, which remained free in solution, while DR2 and the linker formed well-mixed clusters. These data

suggested that DR2 and the linker have the potential to form protein condensates.

We next investigated which specific interactions may drive the disordered regions to cluster. In protein condensates, the contacts formed inside a single polypeptide chain often resemble the ones formed across different polypeptide chains (Tesei et al, 2021),

indicating that the same interactions promote the internal and intermolecular organization. Therefore, we computed contact maps for interactions formed within single polypeptide chains and across different polypeptide chains in the clusters. Indeed, 1-D plots derived from the contact matrices, where we summed up the contributions from all possible interactions of a single residue in the simulations, confirmed this feature. Interactions within and between DR2 and the linker region were mainly formed by the charged and aromatic residues (Asp, Lys, Phe) (Appendix Fig. 3A–D). In DR2, Asp328 and Lys349 formed the most probable contacts, suggesting an important role in cluster formation (Appendix Fig. 3A–D). The contact analysis showed that distinct regions in the disordered segments in IRE1α LD have the propensity to form low-affinity transient interactions driven by aromatic and charged residues. In summary, MD simulations revealed the biochemical potential of the disordered segments in IRE1's LD in driving its LLPS.

## IRE1α cLD forms rigid condensates

As MD simulations predicted that DR2 and the linker region form clusters in isolation, we next tested their LLPS potential through DIC microscopy with the purified constructs. We found that the core lumenal domain (cLD aa 24–389), which lacks the disordered linker region, formed condensates (Fig. 3F, right). These data revealed that the linker region is not necessary for the formation of IRE1α LD condensates. IRE1α cLD rapidly formed condensates at lower protein and PEG concentrations compared to IRE1α LD (Fig. EV2A,B vs Fig. EV3A,B). While the size of IRE1α cLD condensates was highly comparable to that of IRE1α LD, DIC microscopy revealed that IRE1α cLD (aa 24–389) formed structures resembling beads on a string suggesting that they are incapable of fusion (Figs. 3F, right and EV3C). IRE1α cLD condensates accumulated on the glass slide without wetting the surface (Fig. EV3D). FRAP experiments showed that IRE1α cLD recovered after $t_{1/2} = 281.5$ s and displayed a 27.6% mobile fraction confirming that cLD condensates are less dynamic in comparison to IRE1α LD condensates (Fig. 3G; Appendix Table S4). In support of the low mobile fraction of IRE1α cLD shown by the FRAP data, mCherry-tagged IRE1α LD partitioned into preformed IRE1α LD condensates after 5 min, in contrast mCherry tagged-cLD failed to do so (Fig. EV3E). Altogether, our data showed that IRE1α cLD formed stable condensates, indicating that the linker segment (aa 390–443) modulates both the propensity to coalesce and IRE1α LD associations in condensates. Together with the MD simulations, these data suggest that the linker region forms transient intra- and intermolecular contacts with the disordered segments. We anticipate that the linker binding may compete for low-affinity interactions that are involved in IRE1α LD self-assembly leading to an increase in its dynamics and clustering threshold. Altogether, our data reveal that disordered segments potentially regulate a critical switch in UPR signaling.

## Mutations in IRE1α LD's disordered regions modulate clustering in vitro

We next screened for mutants in the disordered segments that may regulate IRE1α LD's clustering. For these experiments, we used LLPS assays in solution to rapidly screen for mutants that impair

IRE1α LD self-assembly. As IRE1α cLD could readily form condensates in solution, we introduced mutations in DR2 and DR3 of IRE1α LD. Based on the MD simulations and published work on LLPS, we chose regions enriched in hydrophobic, aromatic or charged sequences, which might form intermolecular contacts to nucleate phase separation (Choi et al, 2020). Specifically, we mutated 3- or 4-residue stretches to glycine-serine residues, which often form dynamic segments acting as spacers in biomolecular condensates (Fig. 4A).

All the mutants were biochemically stable upon purification, allowing us to study their clustering behavior by DIC microscopy. The IRE1α LD [312]TLPL[315] mutant formed smaller condensates with slower kinetics that failed to fuse efficiently suggesting that this region is important for condensate formation. A mutation in the [320]QTDG[323] segment did not impair phase separation (Figs. 4B and EV4A), whereas, impairing electrostatic interactions predicted by the MD simulations in [346]LKSK[349] largely abolished phase separation (Figs. 4B and EV4B). Yet, mutating the neighboring segment ([350]NKLN[353]) only slightly impacted LLPS (Fig. 4B). Complementarily, mutating a segment enriched in hydrophobic and aromatic residues (IRE1α LD [352]LNYL[355]) impaired the formation of IRE1α LD condensates and introducing a mutation to the following segment [354]YLR[356] substantially compromised LLPS (Fig. EV4C). In contrast, the IRE1α LD [373]TKML[376] mutant in DR3 did not impact condensate formation (Fig. 4B). These data revealed that the [354]YL[355] region forms a hot spot for molecular interactions driving IRE1α LD clustering and both electrostatic and hydrophobic interactions contribute to LLPS of IRE1α LD.

Notably, [354]YL[355] resides near [359]WLLI[362], whose mutation to GSGS impairs the formation of IRE1α LD oligomers (Karagoz et al, 2017; Belyy et al, 2022). DIC microscopy indicated that the IRE1α LD [359]WLLI[362] mutant underwent LLPS, even under conditions in which wild type IRE1α LD barely formed condensates (Figs. 4B and EV4D). These results revealed that disrupting the canonical oligomerization interface did not hinder phase separation of IRE1α LD and, moreover, suggested that oligomers are distinct from condensates (Figs. 4B and EV4D). These results motivated us to interrogate the oligomerization behavior of the IRE1α LD [312]TLPL[315] and [352]LNYL[355] mutants using orthogonal methods. To this end, we performed analytical ultracentrifugation sedimentation velocity (AUC-SV) experiments to determine whether IRE1α LD mutants could form high-order oligomers. These experiments revealed that, similar to previous observations on IRE1α cLD (Karagoz et al, 2017), IRE1α LD was found in equilibrium of dimers and high-order oligomers at 25 μM. Strikingly, under those conditions, IRE1α LD [352]LNYL[355] and [312]TLPL[315] mutants only formed dimers. These data revealed that [352]LNYL[355] and [312]TLPL[315] regions are important for the formation of high-order IRE1α LD oligomers (Fig. 4C).

Tethering a protein to model membranes might impact its clustering propensity compared to in solution due to the restricted conformational freedom or due to limited diffusion to two dimensions. Therefore, we next assessed whether the [352]LNYL[355] and [312]TLPL[315] mutants could form clusters on SLBs. Similar to wild type IRE1α LD, we tethered mCherry-tagged [312]TLPL[315] and [352]LNYL[355] mutants via a 10His-tag on their C-termini to SLBs containing 1% Ni-NTA lipids. To determine the molecular mass and the oligomeric status of mCherry-IRE1α LD-10His and its mutants on SLBs, we used mass photometry (Foley et al, 2021;

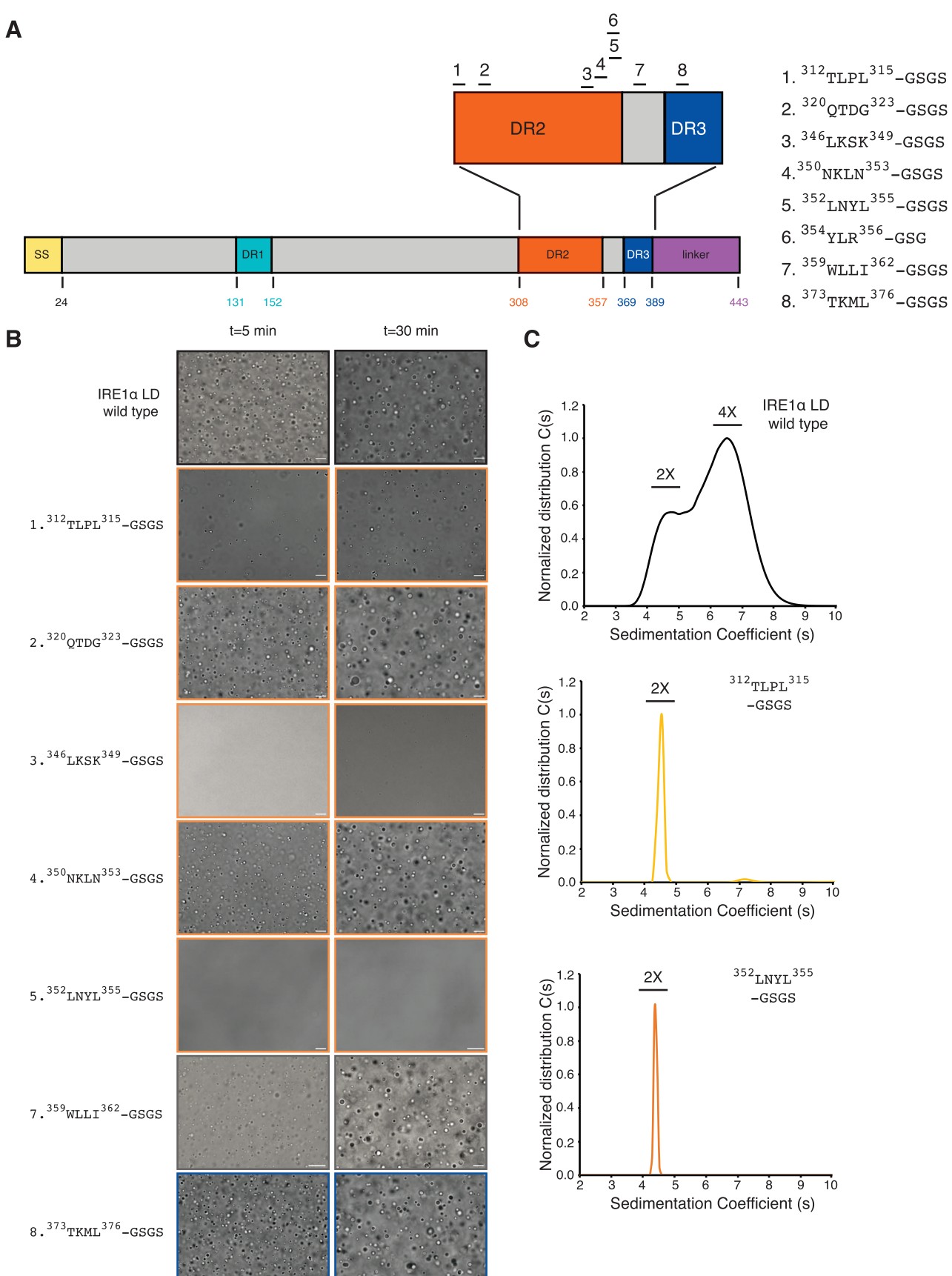

**Figure 4.  Mutations in disordered segments in IRE1α LD impair phase separation.**

(A) Schematic description of the mutations (1–8) introduced to IRE1α LD. SS Signal sequence, DR disordered region. (B) DIC images showing LLPS behavior of IRE1α LD wild type and the mutants at 50 μM after their incubation with 6% PEG for 5 min (left) and 30 min (right). Scale bar = 10 μm. (C) Analytical ultracentrifugation sedimentation velocity curves of 25 μM wild type IRE1α LD (top), IRE1α LD ³¹²TLPL³¹⁵⁻GSGS (middle) and IRE1α LD ³⁵²LNYL³⁵⁵⁻GSGS (bottom) mutants.

Steiert et al, 2022). Mass photometry determines the protein mass by measuring light scattering of single molecules (Sonn-Segev et al, 2020). We performed mass photometry analyses of mCherry-IRE1α LD-10His and its mutants at protein concentrations with optimal particle density to enable single-molecule measurements (between 30 and 100 nM). Up to 50 nM, mCherry-IRE1α LD-10His and its mutants (D123P, ³¹²TLPL³¹, ³⁵²LNYL³⁵⁵) predominantly formed monomers. At 75 nM, the mCherry-IRE1α LD-10His and ³¹²TLPL³¹, ³⁵²LNYL³⁵⁵ started forming dimers on SLBs (Fig. 5A). In contrast, the D123P mutant was mainly monomeric at this condition (Mw= 95 ± 37 kDa) (Fig. 5A). At 100 nM, the mCherry-IRE1α LD-10His and the ³¹²TLPL³¹ mutant were mainly dimeric (Mw wild Type= 135 ± 67 kDa and Mw TLPL = 157 ± 73 kDa), but showed a broad size distribution spanning higher molecular masses, suggesting that the proteins begin to form oligomers at this concentration (Fig. 5A). Notably, at 100 nM, the ³⁵²LNYL³⁵⁵ mutant showed a narrower size distribution with the predicted mass of dimers (Mw= 147 ± 57 kDa). These data were consistent with the in solution LLPS assays, which showed that the ³⁵²LNYL³⁵⁵ mutant was more impaired in condensate formation compared to the ³¹²TLPL³¹ mutant. In agreement with published work (Zhou et al, 2006), at a concentration of 100 nM in solution, mCherry-IRE1α LD-10His and all mutants were monomeric (Fig. 5B). Overall, mass photometry analyses revealed that restriction of these proteins on membranes increased their propensity to self-associate compared to the in solution experiments.

We next tested the clustering efficiency of the mutants upon molecular crowding at various protein concentrations ranging from 100 to 200 nM, including lower concentrations (100 nM) where they mainly formed dimers determined by the mass photometry analyses. In the absence of crowding agent, similar to mCherry-IRE1α LD-10His, the ³¹²TLPL³¹⁵ and ³⁵²LNYL³⁵⁵ mutants displayed diffuse signal and dynamic behavior on SLBs (Fig. EV4E–G). At 100 nM, upon molecular crowding (11% PEG), mCherry-IRE1α LD-10His and its mutants (³¹²TLPL³¹⁵ and ³⁵²LNYL³⁵⁵) started to assemble into clusters shown by TIRF imaging (Figs. 5C and EV4H–J). However, the clusters became more prominent in both intensity and number at higher protein concentrations (Figs. 5C and EV4H–J). Intensity plot analyses of the mCherry signal of the TIRF images revealed that at 150 nM, the mCherry-IRE1α LD-10His displayed a higher number of high intensity clusters compared to the ³¹²TLPL³¹⁵ and ³⁵²LNYL³⁵⁵ mutants suggesting that the wild type is more efficient in cluster formation compared to the mutants at this concentration (Fig. EV4H–J). Importantly, at 200 nM in the presence of 11% PEG, the mutants formed clusters with similar intensity as the wild type IRE1α LD (Figs. 5C and 4H–J, bottom row). Multivalent low-affinity interactions have been shown to drive LLPS of biological systems (Alberti and Dormann, 2019). Our data indicate that on model membranes in the presence of molecular crowding and high protein concentrations, IRE1α LD self-association is favored. Under those conditions, alternative low-affinity interaction networks

could form and drive clustering of IRE1α LD mutants in contrast to the experiments conducted in solution. Under those conditions, the monomerization mutant D123P did not efficiently assemble into clusters, further underling that dimerization is important for building valency on membranes similar to what we observed in solution (Figs. 5D and EV4K). Altogether, our data converge on a model in which IRE1α LD dimers interact with each other in various conformations. These interactions are facilitated by the DRs driving low-affinity contacts with no fixed valence to allow the assembly of IRE1α LD into stable clusters on membranes. We propose that the transient low-affinity interactions are crucial in bringing IRE1α molecules in close proximity to drive the formation of active IRE1α assemblies.

## The disordered regions in IRE1α LD are important for its clustering and signaling in cells

IRE1α forms foci in cells experiencing ER stress (Li et al, 2010; Ricci et al, 2019; Belyy et al, 2020; Tran et al, 2021). To assess the role of the disordered segments in IRE1α LD clustering in cells, we established stable cell lines expressing wild type human IRE1α or IRE1α mutants with impaired (IRE1α LD ³¹²TLPL³¹⁵ and ³⁵²LNYL³⁵⁵ mutants) or enhanced (IRE1α cLD Δlinker) clustering, as determined by our in vitro assays (Figs. 3F and 4B). Moreover, we generated cell lines expressing the previously established IRE1α LD dimerization (D123P) and oligomerization (³⁵⁹WLLI³⁶²) mutants (Zhou et al, 2006; Kitai et al, 2013; Karagoz et al, 2017; Belyy et al, 2022). We introduced doxycycline-inducible transgenes encoding mNeonGreen (mNG) tagged variants of IRE1α into mouse embryonic fibroblasts (MEFs) deficient for both isoforms of IRE1 (IRE1 α⁻/⁻ and IRE1β⁻/⁻) and monitored IRE1α clustering by fluorescence microscopy. We introduced the mNG tag into IRE1α's cytoplasmic flexible linker (Li et al, 2010; Karagoz et al, 2017; Belyy et al, 2020). In the absence of doxycycline, cells expressed low levels of IRE1α due to the inherent leakiness of the doxycycline-inducible system (Appendix Fig. 4A–E). Under these conditions, the expression levels of IRE1α-mNG and its mutant variants were similar to the level of endogenous IRE1α observed in wild type MEFs as assessed by Western blot analysis (Appendix Fig. 4D,E). When we treated the cells with the ER stress inducing drug tunicamycin, cells carrying IRE1α-mNG showed a modest reduction in XBP1 mRNA splicing activity compared to wild type control MEFs, suggesting that the mNG-tag slightly impairs its activity (Appendix Fig. 4F).

The size of IRE1α clusters in the cell depends on the protein concentration (Li et al, 2010; Belyy et al, 2022; Gomez-Puerta et al, 2022). While in most tissues endogenous IRE1α clusters are generally too small to overcome the diffraction limit of light, IRE1α forms microscopically visible clusters observed as distinct foci when it is ectopically expressed to levels 2–20 times over endogenous protein levels (Li et al, 2010, Karagoz et al, 2017; Ricci et al, 2019; Belyy et al, 2020; Tran et al, 2021; Belyy et al, 2022; Gomez-Puerta

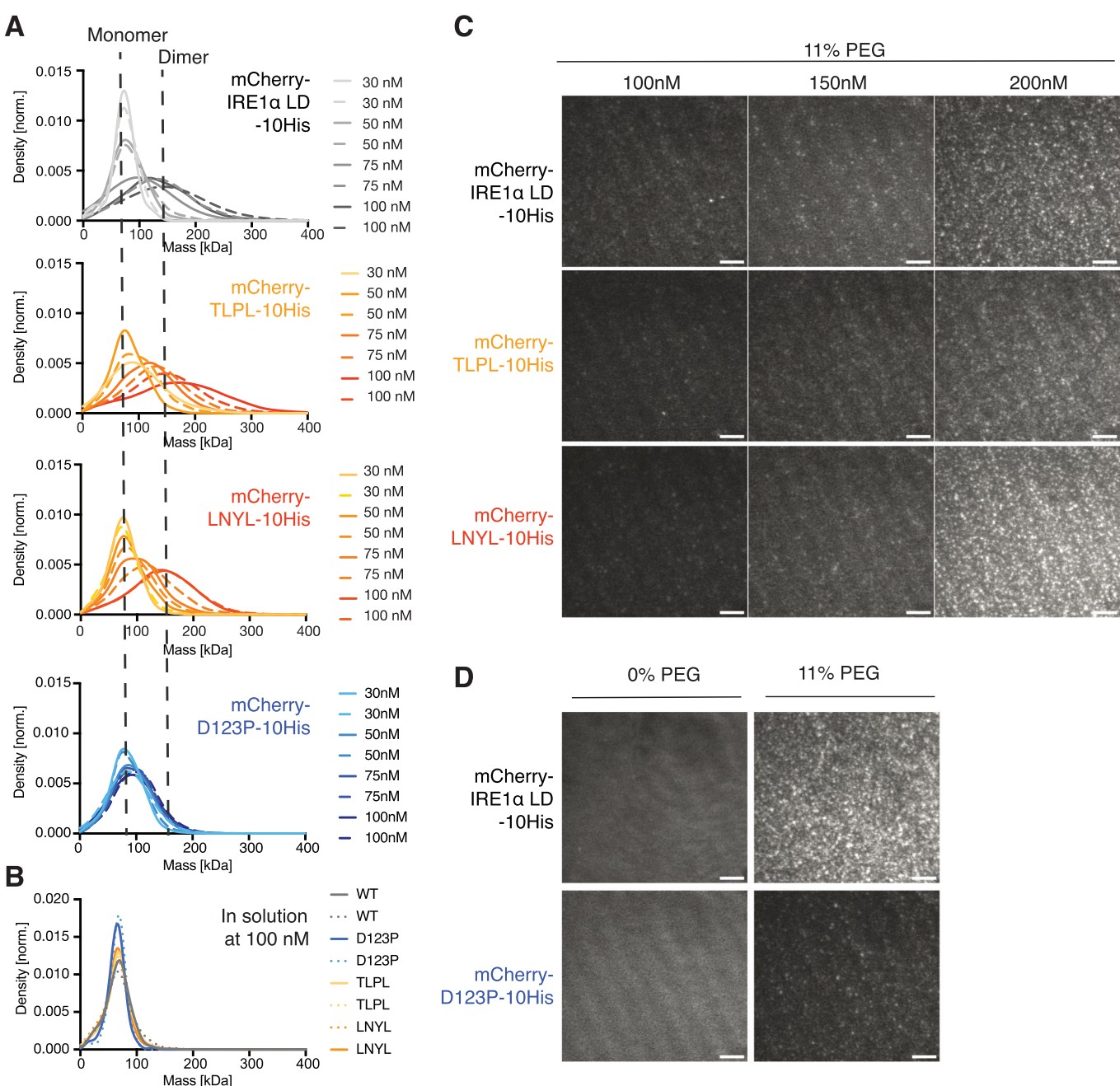

**Figure 5.  Membrane-tethering facilitates IRE1α LD self-assembly.**

(A) Size distribution of mCherry-IRE1α LD-10His, [312]TLPL[315]-GSGS, [352]LNYL[355]-GSGS and D123P mutants on SLBs measured by mass photometry experiments. The lines and the dashed lines show experimental replicates at different protein concentrations coupled to SLBs. (B) Size distribution of mCherry-IRE1α LD-10His, [312]TLPL[315]-GSGS, [352]LNYL[355]-GSGS and D123P mutants in solution at 100 nM measured by mass photometry experiments. The lines and the dashed lines show experimental replicates protein concentration. (C) TIRF images of mCherry-IRE1α LD-10His and the [312]TLPL[315]-GSGS, [352]LNYL[355]-GSGS mutants on SLBs in the presence of 11% PEG at different protein concentrations (100–200 nM). Scale bar = 5 μm. (D) TIRF images of mCherry-IRE1α LD-10His and D123P mutant on SLBs in the presence of 11% PEG at 200 nM. Scale bar (SB) = 5 μm.

et al, 2022). To visualize IRE1α clustering in mammalian cells with confocal microscopy, we overexpressed IRE1α in MEFs in a controlled manner using a doxycycline-inducible promoter. IRE1α-mNG expression levels increased linearly with doxycycline concentration in the range of 25–400 nM (Appendix Fig. 4C). Treatment with 400 nM doxycycline led to expression levels of

IRE1α-mNG and its mutants roughly 30-fold over endogenous IRE1α in wild type MEFs (Appendix Fig. 4C,D,G). Confocal microscopy experiments in which we monitored IRE1α cluster formation indicated that at 400 nM doxycycline, but not 100 nM, cells expressing wild type IRE1α-mNG formed microscopically visible foci in a stress-dependent manner (Figs. 6A and EV5A,C).

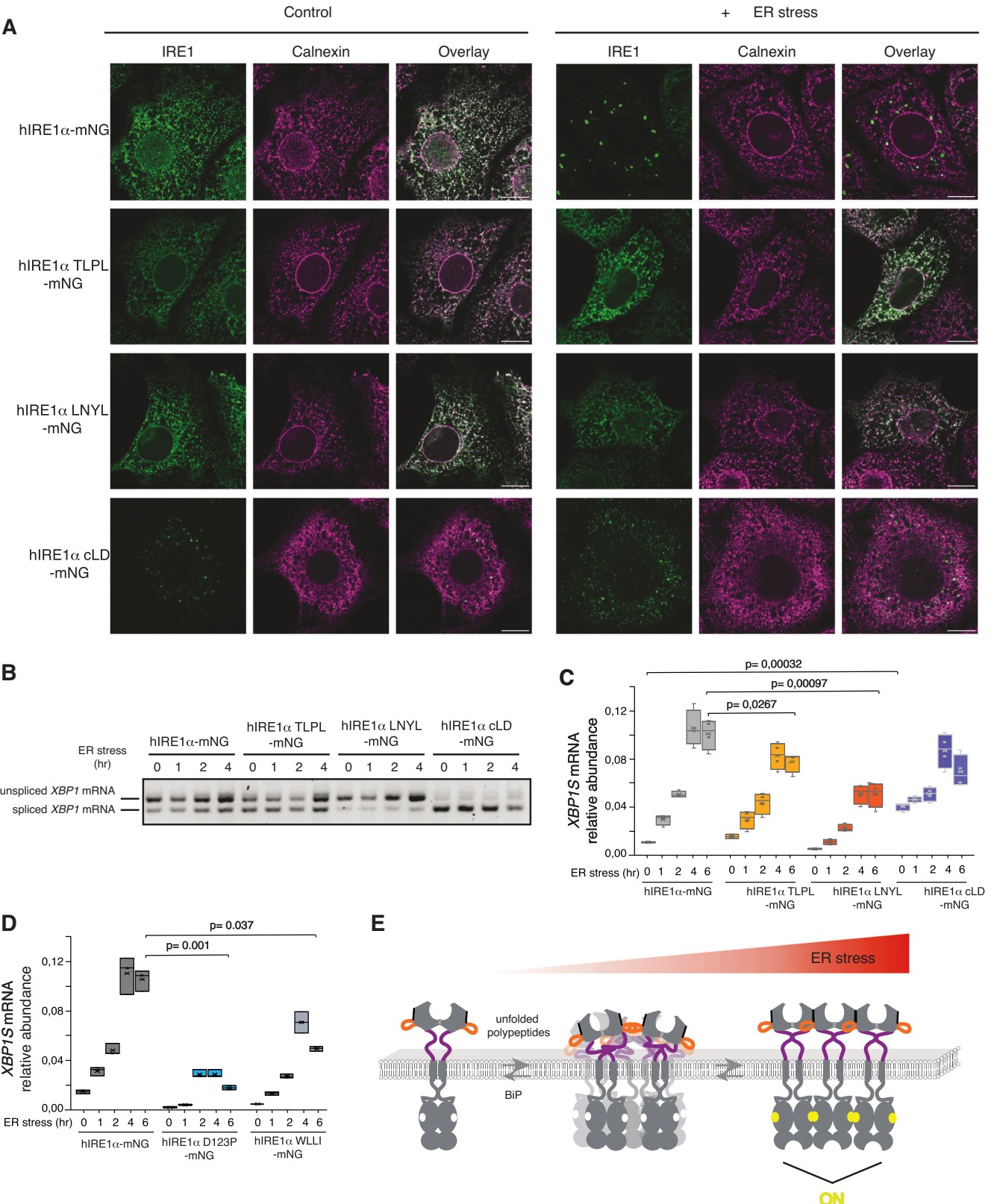

◄ **Figure 6. IRE1α LD DR mutants dysregulate its clustering and activity in vivo.**

(A) Immunofluorescence images of MEFs treated with 400 nM doxycycline expressing IRE1α-mNG or its mutants in the absence (left panel) of stress and treated with 5 μg/ml ER stressor Tunicamycin for 4 h (right panel). IRE1α-mNG and its mutants are visualized by mNG fluorescence (green) and the ER-chaperone Calnexin is stained by anti-calnexin antibody (purple). Scale bar = 10 μm. (B) Semi-quantitative PCR reaction to monitor splicing of *XBP1* mRNA by IRE1α-mNG and its mutants at different time points after induction of ER stress by addition of 5 μg/ml Tunicamycin. The expression of the IRE1α variants is induced by a 24-h treatment of MEFs with 400 nM doxycycline before induction of ER stress. The bands are indicated as unspliced and spliced *XBP1* variants. (C) qRT-PCR to monitor splicing of *XBP1* mRNA by IRE1α-mNG and its mutants at different time points after induction of ER stress by the addition of 5 μg/ml Tunicamycin. The expression of the IRE1α variants is induced by a 24-h treatment of MEFs with 400 nM doxycycline before induction of ER stress. Displayed are the values of $n = 4$ independent experiments resulting in the displayed *P* value. The *P* values were determined by the two-sided Student's *t* test. The data are shown as box plots. The central line in the box plot marks the median and the cross shows the mean, the boxes mark the first and third quartiles and the whiskers display the maximum and the minimum points. (D) qRT-PCR to monitor splicing of *XBP1* mRNA by IRE1α-mNG, the dimerization mutant (IRE1α-D123P-mNG) and the oligomerization mutant (IRE1α-WLLI-mNG) at different time points after induction of ER stress by the addition of 5 μg/ml Tunicamycin. The expression of the IRE1α variants is induced by a 24-h treatment of MEFs with 400 nM doxycycline before induction of ER stress. Shown are the values from $n = 3$ independent experiments resulting in the displayed *P* value. The *P* values were determined by a two-sided Student's *t* test. The data are shown as box plots. The central line in the box plot marks the median and the cross shows the mean, the boxes mark the first and third quartiles and the whiskers display the maximum and the minimum points. (E) Model describing the role of DRs in IRE1α clustering. During ER stress, the ER-resident chaperone BiP is released from the DRs in IRE1α LD allowing these segments to self-associate through multivalent weak interactions. Concurrently, direct binding of misfolded proteins accumulating in the ER enhances the formation of dynamic IRE1α assemblies through additional multivalent interactions. These dynamic assemblies efficiently transition into stable IRE1α clusters with distinct signaling competent interfaces (light grey: dimerization interface, black: oligomerization interface) and conformation allowing for IRE1α trans-autophoshorylation and RNase activity.

Consistent with published work, at 400 nM Dox, IRE1α-mNG showed low constitutive *XBP1* mRNA splicing activity due to overexpression, which further increased upon treatment with tunicamycin (Fig. 6B) (Li et al, 2010).

Consistent with published work, the IRE1α-WLLI-mNG mutant did not form foci under ER stress conditions and displayed a diffuse signal in the ER under overexpression conditions (Fig. EV5C) (Karagoz et al, 2017; Belyy et al, 2022). Under the same conditions, the IRE1α-TLPL-mNG and IRE1α-LNYL-mNG mutants failed to form foci at comparable expression levels as the wild type IRE1α-mNG (Fig. 6A; Appendix Fig. 4A,D). These results indicate that the TLPL and LNYL mutants are defective in clustering in cells compared to wild type IRE1α-mNG. These results substantiated our findings that disordered segments in the LD are important for IRE1α assembly. In stark contrast, IRE1α cLD-mNG (Δlinker), which lacks the linker region, formed clusters constitutively in the absence of stress (Fig. 6A), or even upon induction of its expression with a lower (100 nM) doxycycline concentration (Fig. EV5A). This suggested that the clustering threshold is lower for IRE1α cLD-mNG consistent with the results we obtained in our in vitro experiments. Moreover, the clusters formed by IRE1α cLD-mNG (Δlinker) were smaller compared to the IRE1α-mNG. We speculate that the formation of highly rigid condensates by IRE1α cLD observed in vitro might lead to impaired fusion and consequently the formation of smaller clusters observed in cells. Taken together, these results corroborate the role of disordered segments in IRE1α's LD in regulating IRE1α's self-association into high-order assemblies.

Next, we investigated whether the mutants would impact IRE1α activity monitored by its ability to splice *XBP1* mRNA. Semi-quantitative polymerase chain reaction (PCR) and quantitative real-time PCR (qRT-PCR) analyses revealed that IRE1α LNYL-mNG mutant exhibited impaired *XBP1* mRNA splicing activity compared to wild type IRE1α-mNG, indicating that the interfaces formed by the LNYL region in IRE1α LD play an important role in forming active IRE1α assemblies in cells (Fig. 6B,C). Instead, the IRE1α TLPL-mNG splicing activity was slightly but consistently diminished, where the TLPL mutant did not reach the same splicing efficiency as IRE1α-mNG (Belyy et al, 2022; Gomez-Puerta

et al, 2022) (Figs. 6B,C and EV5B; Dataset EV1). Those results did not depend on IRE1α-mNG expression levels as we got similar results upon treatment with 25 nM doxycycline, where IRE1α-mNG and the mutants were expressed around 3-fold over the endogenous IRE1α level (Fig. EV5B; Appendix Fig. 4C,E). These data suggest that in cells, the mutation in the interface formed by the ³¹²TLPL³¹⁵ segment can be largely compensated for by the formation of other protein interfaces, consistent with our in vitro results. Notably, IRE1α cLD-mNG, which constitutively formed foci in the absence of stress, displayed high constitutive *XBP1* mRNA splicing activity independent of IRE1α-mNG expression levels (Figs. 6B,C and EV5B; Appendix Fig. 4B,D,G; Dataset EV1). Under those conditions, the dimerization mutant IRE1α-D123P-mNG was inactive regardless of its expression level (Figs. 6D and EV5D) (Kitai et al, 2013). Similarly, the IRE1α-WLLI-mNG mutant was impaired in *XBP1* mRNA splicing when expressed near endogenous levels (25 nM doxycycline) supporting previous work (Karagoz et al, 2017). At higher expression levels (400 nM doxycycline), the WLLI mutant showed comparable activity to the LNYL mutant (Figs. 6D and EV5D). At these expression levels, the *CHOP* mRNA levels were highly similar suggesting that, under acute ER stress conditions, cells expressing IRE1α-mNG mutants did not show increased apoptotic potential through PERK hyperactivation (Fig. EV5E; Dataset EV1). Altogether, these results indicate that disordered segments in the IRE1α LD mediate the formation of signaling competent IRE1α assemblies in cells.

## Discussion

IRE1 governs the most evolutionarily conserved branch of the UPR. IRE1 signaling is tied to the formation of clusters in yeast and mammalian cells, and mutations that impair IRE1 oligomerization result in severely reduced activity (Aragon et al, 2009; Li et al, 2010; Karagoz et al, 2017; Belyy et al, 2020). Thus, self-assembly emerges as a fundamental principle of IRE1 regulatory control. IRE1 clustering is driven by its sensor LD, which juxtaposes its cytosolic domains to activate its RNase domain. The structural features enabling IRE1α LD clustering and its mechanistic principles have

remained unknown, and here, through bottom-up approaches to reconstitute IRE1α LD clustering in solution and on model membranes, we provide evidence for the role of DRs in regulating IRE1α's self-assembly.

We found that the stress sensing LD of IRE1α formed dynamic biomolecular condensates in solution. In contrast, IRE1α LD formed long-lived clusters on model membranes similar to what was shown for IRE1α clusters in cells indicating that membrane-tethering stabilizes interactions among IRE1α LD molecules (Fig. 1C–E) (Belyy et al, 2020). We anticipate that these long-lived interactions are crucial to provide sufficient time to build up signaling competent interfaces within the LD to allow for the transmission of the information across the membrane bilayer to initiate the autophosphorylation of the kinase domains leading to the activation of its RNase domain. This model is in line with recent data, which showed a lag between IRE1α oligomerization and its trans-autophosphorylation activity in cells (Belyy et al, 2022). In our experiments, we used a simple membrane composition, and thus future studies are necessary to assess how changes in the membrane composition during ER stress might regulate IRE1α clustering (Volmer et al, 2013; Halbleib et al, 2017; Fun and Thibault, 2020).

Clustering of IRE1α LD on membranes followed a sharp transition as a function of molecular crowding (compare 10% and 11% PEG, Fig. 1F), suggesting that an increase in ER protein load, as during ER stress, could constitute the sensing threshold for IRE1α (Zuber et al, 2004; Liu et al, 2005; Rhodes, 2005; Snapp et al, 2006; Liu et al, 2007; Covino et al, 2018). In line with our previous observations (Karagoz et al, 2017), we found that in addition to molecular crowding, IRE1α LD's direct interaction with unfolded peptide ligands decreased the threshold for IRE1α LD clustering (Fig. 1F–H) and stabilized IRE1α LD condensates. We anticipate that misfolded proteins with diverse biochemical properties could differently modulate the threshold for IRE1α clustering and the stability of IRE1α clusters regulating both sensitivity and duration of the UPR in cells.

Biomolecular condensates are formed through multivalent low-affinity interactions by disordered segments in proteins (Patel et al, 2015; Banani et al, 2017; Lin et al, 2017; Alberti and Hyman, 2021). IRE1α LD has several DRs whose function has remained largely unknown. Surprisingly, the distinct DRs in IRE1α LD regulate the formation and dynamics of IRE1α LD condensates in opposite ways. Removal of the linker region (aa 390–443), which connects the folded core domain to the transmembrane helix, decreased the clustering threshold of IRE1α both in vitro and in cells. In contrast, mutating aromatic and hydrophobic amino acids in two different parts of the DR2 segment ([312]TLPL[315] and [352]LNYL[356] mutants) largely impaired IRE1α clustering in vitro and in cells. Using complementary reconstitution approaches, we found that these mutants impair IRE1α LD clustering to a different degree. The AUC-SV measurements revealed that both [312]TLPL[315] and [352]LNYL[356] mutants were abolished in the formation of high-order oligomers in solution (Fig. 4C). However, they showed distinct differences in their propensity to form condensates upon molecular crowding (Fig. 4B), where the [312]TLPL[315] mutant formed small condensates, while the [352]LNYL[356] mutant was largely abolished in condensate formation. Mass photometry data on SLBs confirmed these results showing that the [312]TLPL[315] mutant started to assemble into oligomers in a concentration-dependent manner, whereas the [352]LNYL[356] mutant only formed dimers under these conditions (Fig. 5A). In line with these results, the mutants impacted IRE1α's XBP1 mRNA splicing activity to different extents.

While the [352]LNYL[356] mutation largely abolished XBP1 mRNA splicing activity, the XBP1 mRNA splicing activity of the [312]TLPL[315] mutant was only slightly impaired (Fig. 6B,C). Based on our in vitro results showing that the [312]TLPL[315] mutant was not completely impaired in its ability to form condensates in solution and clusters on membranes, we speculate that the [312]TLPL[315] mutant may form small clusters below the diffraction limit of light in cells allowing for the trans-autophosphorylation of its kinase domains and its activation. This is in line with previous work showing that the formation of small oligomers is sufficient to drive IRE1 activation (Karagoz et al, 2017; Belyy et al, 2022; Gomez-Puerta et al, 2022).

Upon molecular crowding on SLBs, the [312]TLPL[315] and [352]LNYL[356] mutants were less efficient in cluster formation compared to the wild type IRE1α LD at lower protein concentrations. However, at higher protein concentrations under those conditions, both [312]TLPL[315] and [352]LNYL[356] mutants could form clusters on SLBs similar to the wild type IRE1α LD (Fig. 5C). These data indicate that under the conditions that favor protein clustering by increasing the effective protein concentration, namely on membranes and in the presence of molecular crowding, the formation of alternative low-affinity interaction networks by the disordered segments could compensate for the loss of the contacts of the mutated individual segments. We anticipate that the membrane composition or geometry and binding of misfolded polypeptides might allow the establishment of these low-affinity interactions in various physiological and pathological contexts to initiate IRE1α signaling.

The IRE1α LD [359]WLLI[362] mutant does not form oligomers (Karagoz et al, 2017) but could readily form condensates even at lower protein concentrations compared to the wild type IRE1α LD. These data indicate that the distinct oligomeric conformation formed through the contacts provided by the [359]WLLI[362] segment is not required for the multivalent interactions formed by the [312]TLPL[315] and [352]LNYL[356] regions. Both the [359]WLLI[362] and [352]LNYL[356] mutants display impaired XBP1 mRNA splicing activity in mammalian cells indicating that they both contribute to the assembly of IRE1α into enzymatically active clusters. Interestingly, the [359]WLLI[362] mutation has a stronger effect on IRE1α activity when expressed near the endogenous expression level underlining the importance of oligomer formation for IRE1α signaling (Figs. 6C,D and EV5B,D) (Karagoz et al, 2017). Our data suggest that the [312]TLPL[315] and [352]LNYL[356] regions contribute to the formation of distinct IRE1α assembly intermediates to generate signaling competent IRE1α oligomers in cells. Intriguingly, recent cryo correlative light and electron microscopy (cryo-CLEM) imaging of IRE1α clusters in mammalian cells suggested that IRE1α LD forms ordered double-helical filaments in its native, membrane-embedded state under stress conditions (Tran et al, 2021). We anticipate that the elevated local concentration of IRE1α in the clusters might allow for the assembly of IRE1α LD into filaments with the distinct structural arrangement observed in cells. The DRs in IRE1α LD, which we identified to regulate its clustering, were earlier proposed to be recognized by the ER-chaperone BiP (Amin-Wetzel et al, 2019), and therefore, it is plausible that these regions are occluded by BiP binding, which could prevent clustering under non-stress conditions. These results support the novel function of the DRs as regulatory hubs for stress-induced activation of IRE1α. While the DRs in the IRE1 LD are not highly conserved at the amino acid level, these regions have regulatory

functions in budding yeast suggesting their functional conservation (Kimata et al, 2007; Oikawa et al, 2007). Intriguingly, all three UPR sensors have DRs in their ER luminal sensor domains and similar mechanisms might hold true for PERK and ATF6.

Altogether, our findings reveal the regulatory role of DRs in IRE1α's LD in its assembly within cells. We speculate that low-affinity interaction networks formed by the DRs would enable formation of dynamic assembly intermediates with multiple conformations and this provides an advantage for protein assembly on membrane surfaces, where conformational freedom is limited. Around half of all transmembrane proteins carry disordered segments that are 30 amino acids long or more (Burgi et al, 2016), suggesting that DR-controlled clustering might be a common feature among many membrane proteins.

Our data converge on a model in which ER stress triggers BiP release from the DRs in IRE1α's LD, allowing their association with DRs of other IRE1α molecules through multivalent low-affinity contacts. ER stress increases molecular crowding in the ER due to the impairment in protein-folding and secretion and accumulation of unfolded proteins, all of which facilitate IRE1α self-assembly. Under those conditions, unfolded polypeptides that bind to IRE1α LD and membrane-imposed constraints further stabilize IRE1α clusters leading to the formation of stable IRE1α assemblies competent in UPR signaling (Fig. 6E).

Phase separation of membrane-associated and transmembrane proteins has emerged as a novel mechanism regulating subcellular organization and signaling on membranes (Banjade and Rosen, 2014; Su et al, 2016; Case et al, 2019; Ditlev et al, 2019; King et al, 2020; Rebane et al, 2020). Our data suggest that multivalent weak interactions of IRE1α driven by its DRs in its LD contributes to UPR signaling. IRE1α levels are controlled via intricate feedback loops that regulate its abundance during ER stress (Sun et al, 2015) and aberrant overexpression of IRE1α in multiple myeloma and breast cancer contributes to pathology (Harnoss et al, 2019; Harnoss et al, 2020). We anticipate that DR-driven novel assembly states of IRE1α identified here could be altered in pathology and shall guide future studies aimed at targeting IRE1α for therapeutic purposes in disease.

# Methods

## Generation of constructs for in-vitro assays

All constructs were constructed in pET-47 b(+) vector. hIRE1α LD mutants [312]TLPL[315]-GSGS, [350]NKLN[353]-GSGS, and [352]LNYL[355]-GSGS were based on the T274C variant of hIRE1α LD (Cysteines are substituted by Alanines, Threonine aa274 was substituted by Cysteine). All other mutants were based on the WT hIRE1α LD. We could not observe any differences between hIRE1α LD WT and T274C in our assays. [320]QTDG[323]-GSGS and [373]TKML[376]-GSGS were constructed through site-directed mutagenesis, with subsequent blunt end ligation. For mCherry tagged proteins, an N-terminal mCherry sequence and C-terminal 10HisTag was used.

## Protein expression and purification

hIRE1α LD expression and purification were adapted from published protocols (Karagoz et al, 2017). In brief, *Escherichia coli* strain BL21DE3* RIPL was grown with the respective antibiotics in Luria Broth at 37 °C until $OD_{600}$ = 0.6–0.8. The protein expression was induced with 400 μM IPTG for hIRE1α variants without and 1 mM for variants with mCherry at 20 °C and grew overnight. Before lysis and after each purification step, 1× Roche cOmplete Protease Inhibitor Cocktail was added to the cells or fractions containing protein. Cells were harvested and lysed (50 mM HEPES pH 7.2–7.4, 400 mM NaCl, 20 mM Imidazol, 5 mM β-mercaptoethanol, 0 to 10% Glycerol) in an Avestin EmulsiFlex-C3 cell disruptor at 16,000 psi. The lysate was spun at 30,700 × *g* for 45 min. The supernatant was applied to a 5 ml His-TRAP column (GE Healthcare) and eluted with a gradient of 20 mM to 500 mM imidazole. The eluate was diluted with 50 mM HEPES pH 7.2–7.4 (in the absence or in the presence of 5 or 10% Glycerol, 5 mM β-mercaptoethanol) to a concentration of 50 mM NaCl, to apply it to a HiTRAP Q HP (5 ml, GE Healthcare) anion exchange column. The protein was eluted with a linear gradient from 50 mM to 1 M NaCl. To remove the His tag from hIRE1α LD without mCherry, the protein was incubated with 3C Precision protease at a ratio of 50 to 1 over night at 4 °C. The protein was loaded to a His-TRAP column before it was further purified on a Superdex 200 10/300 gel filtration column (25/50 mM HEPES pH 7.2–7.4, 150 mM NaCl, 5 mM DTT; for mCherry proteins: 25 mM HEPES pH 7.4, 150 mM KCl, 10 mM MgCl$_2$, 5 mM DTT). The Expasy ProtParam tool (http://web.expasy.org/protparam/) was used to determine the extinction coefficient at 280 nm to get the final protein concentration. All the expression plasmids for the proteins are available upon request.

## SLB preparation and assays

The protocol was adapted from (Bakalar et al, 2018; Cell (Bakalar et al, 2018)).

### SUV preparation

In brief, SUVs were prepared by creating a dried lipid film of mainly POPC (1-palmitoyl-2-oleoyl-sn-glycero-3-phosphocholine, Avanti), 1 mol% Ni-NTA (DGS-NTA(Ni) (1,2-dioleoyl-sn-glycero-3-[(N-(5-amino-1-carboxypentyl)iminodiacetic acid)succinyl] (nickel salt)), Avanti) and 0.08 mol% Atto488 labeled DPPE (1,2-Dipalmitoyl-sn-glycero-3-phosphoethanolamine labeled with Atto488, Sigma-Aldrich) with an argon stream followed by desiccation for 45 min. The rehydration was performed with deionized water by gently vortexing followed by 40 s tip sonication at 20% power for three times with 20 s in between to prevent generation of heat. The SUVs were filtered through a 0.22 μm PES filter (Carl Roth) and stored at 4 °C for a maximum of 48 h to prevent oxidation of lipids.

### SLB preparation

SLBs were formed in a silicone chamber (Grace BioLabs, GBL103280) sealed on an RCA cleaned 1.5 H, 24 ×50 mm coverslip (Carl Roth) by fusing 20 μl SUVs with 30 μl MOPS buffer (25 mM MOPS pH 7.4, 125 mM NaCl) for 10 min at room temperature. The SLB was washed with 50 μl PBS and 50 μl wash buffer (25 mM HEPES, pH 7.3, 150 mM NaCl, 250 μM TCEP) each four times, respectively. To determine the optimal concentration of protein in our reconstituted system, we incubated SLBs with various concentrations of mCherry-IRE1α LD-10His ranging from 50 nM to 500 nM, and set on a concentration of 200 nM, which was below saturation of the 1 mol% Ni-NTA lipids as determined by

fluorescent intensity as a function of protein concentration (Appendix Table S1). The mCherry tagged mutants were compared to mCherry-IRE1α LD-10His at 100 nM, 150 nM and 200 nM concentration.

The protein was attached to the membrane by incubating for 10 min followed by three more wash steps with wash buffer to remove any unattached protein from the solution. Imaging was conducted on an Olympus cellSens Live Imaging TIRF system with an Olympus 100 × 1.49 NA high-performance TIRF objective with 7% 488, 100 ms exposure and 10% 561, 100 ms exposure via a Hamamatsu ImagEM X2 EM-CCD camera operated by Olympus cellSens 3.1.1. The fluidity of the membrane was confirmed via FRAP experiments. A 2 s 50% single-point laser pulse of 405 nm was used to bleach the fluorescence of the membrane and protein and the fluorescence recovery was followed over 100 frames every 2 s. Image processing was performed in ImageJ by selecting the FRAP ROI and another ROI of the same size on a non-FRAPed area as bleaching background and was kept the same within an experiment. The bleaching ROI was used to obtain bleaching factors by which the FRAP values were corrected with, followed by normalization. The normalized FRAP values of all 100 frames for mCherry proteins and 15 frames for Atto488 labeled DPPE was fitted to an exponential recovery with no offset curve in ImageJ. The half-life time was used to calculate the diffusion coefficient based on Axelrod et al and Soumpasis et al (Axelrod et al, 1976; Soumpasis, 1983) assuming only 2D diffusion of the protein on the SLBs as any access unbound protein was washed out. Image processing was performed in ImageJ adjusting the brightness and contrast of the images to be the same within a Figure panel.

The intensity plots in Fig. EV5D–G were generated in ImageJ by drawing three lines at identical position on three images from different field of views collected in the same experiment. The values were exported and plotted in PRISM.

### Crowding assay in 2D

The protein of interest was incubated with the desired PEG concentration in 25 mM HEPES pH 7.3, 150 mM NaCl, 250 μM TCEP and PEG (indicated in the Figure legend) for 10 min before FRAP experiments were performed to access the dynamics of the membrane and the protein. For the wash-out experiments, the well was washed five times with 30 μl wash buffer before another FRAP experiment was performed.

### Peptide experiments on SLBs

After carefully washing the access protein, the peptides were incubated for 30 min to allow for binding to mCherry-hIRE1α LD-10His. Phase separation was induced with 25 mM HEPES pH 7.3, 150 mM NaCl, 250 μM TCEP and PEG at percentages between 9% and 11%. After a 10 min incubation period, the Atto488 labeled membrane and the mCherry tagged protein were imaged. The end concentration of the peptides above the SLB were 10 μM MPZ1N, 1 μM MPZ1N -2X and 1 μM MPZ1N -2X-RD.

## Mass photometry on SLBs

### Glass preparation

Glass slides of 1.5 H, 22 ×22 mm and 1.5 H, 24 ×50 mm (VWR) were cleaned in sequential 5 min sonication in Milli-Q water, isopropanol, Milli-Q water, isopropanol and Milli-Q water. The

slides were dried using compressed air and activated for 30 s in a Zepto plasma cleaner (Diener electronics) at maximum power. The smaller slide was attached to the larger slide using double-sided sticky tape (Tesa) resulting in a flow chamber.

### SLB formation SLB within the flow chamber

SUVs of 99 mol% POPC and 1 mol% Ni-NTA at a total concentration of 5 mM were prepared as described above and rehydrated in buffer of 200 mM KCl, 50 mM Tris at pH 7.4. After tip sonicating and filtering of the SUVs, they were mixed in a ratio of 1 to 10 with a buffer containing 150 mM KCl, 50 mM Tris, 5 mM MgCl$_2$ and 2 mM CaCl$_2$ at pH 7.4 to allow for SLB formation in the assembled flow chamber for 30 min at 37 °C. Rigorous washing with wash buffer (25 mM HEPES, pH 7.3, 150 mM NaCl, 250 μM TCEP) allowed for the removal of unfused vesicles.

### Mass calibration

To be able to correlate the interferometric scattering contrast to a known mass, the known assemblies of BSA (Sigma-Aldrich) were used to create a calibration curve at the beginning of every experiment. Therefore, 20 μl of 500 nM BSA in a silicone chamber (3 mm × 1 mm, Grace Bio Labs) on a 1.5 H, 24 ×50 mm were recorded for 3 min in advanced and custom mode in solution. In order to set up membrane experiments, the frame rate was set to 999 Hz with a binning of 5 while the exposure time was set to 0.95 ms. The width and height were set to 512 px and 140 px, respectively, with a horizontal offset of 648 px and a vertical offset of 435 px and a binning of 4. The Acousto-Optic Deflectors were set to a frequency of 50.9990 kHz and 78.430 kHz in x and y, respectively. The Offset Center for both was set to 2.5 V, the Offset Idle to 0.6 V and the Amplitude to 0.3 V while the runtime was set to 0.998 ms. The amplifier voltage was set to 5 V for all image series. The monomer, dimer and trimer were assigned to their known mass in the DiscoverMP software (v2023 R1.2, Refeyn Ltd).

### Mass measurement

The highest concentration of 100 nM that was used on SLBs was used to record the proteins in solution similar to the calibration curve. On SLBs, 50 μl of the desired protein was added sequentially. In total, 50 nM were reached in 10 nM steps followed by two further incubation steps with 25 nM to reach a total concentration of 100 nM ($n = 2$ flow chambers per protein). At every step, the protein was allowed to attach to the Ni-NTA lipids within 5 min incubation time. At 30 nM, 50 nM, 75 nM and 100 nM two different regions were recorded for 3 min in advanced and costum mode with the settings described in the mass calibration section. Analysis was done using the Python script provided in Steiert et al (Steiert et al, 2022). In brief, the videos from the spectrometer were visualized and a threshold set on a membrane without protein to enable precise particle tracking. The calibration parameters and imaging settings were fed into the script and the 2-dimensional (2D) maps of the diffusion coefficient and mass were generated. We extracted the marginal probability distribution of the molecular mass to represent the mass shifts. These curves were fitted to a non-linear regression fit with a Gaussian distribution in PRISM.

## In solution phase separation assay

The phase separation behavior of hIRE1α LD protein variants in presence of PEG-8000 (Sigma-Aldrich, P2139), 40% (w/w)

PEG-8000 (Sigma-Aldrich, P1458) or Ficoll-400 (Sigma-Aldrich, F4375) were observed via DIC microscopy on a Zeiss Axio Observer inverted microscope. Images were acquired at room temperature with a Plan-Apochromat 63×/1.4 Oil DIC RMS objective and CoolSnapHQ2 or Hamamatsu ORCA-Flash4.0 LT+ Digital CMOS camera controlled by Visitron and Zeiss systems, respectively. Therefore, glass wells (Greiner Bio-One 96 Well SensoPlate™) were pretreated with 1% (w/v) Pluronic® F-127 (PF127, Sigma-Aldrich) for 2 h at room temperature. After three wash steps (150 mM NaCl, 1 M HEPES pH 7.3 (Molecular Biology, Fisher BioReagents™), phase separation of the protein of interest was induced. Hence, the protein was mixed with equal volumes of PEG containing buffer (25 mM HEPES pH 7.3, 150 mM NaCl, 4 mM DTT, 20 mM $MgCl_2$, 2-times of final PEG or Ficoll concentration depending on the condition) in a final volume of 50 µl. For peptide experiments, 24.5 µM hIRE1α LD and 0.5 µM mCherry-hIRE1α LD-10His were preincubated with the respective peptide for 30 min on ice before phase separation was induced via PEG. The final protein and PEG concentration and incubation time is indicated in the Figure legends. Image processing was performed in ImageJ and Adobe Photoshop® adjusting the brightness, contrast and sharpness of the images.

## FRAP on condensates in solution

Phase separation was induced as described in the "In solution phase separation" section. For FRAP experiments hIRE1α LD was mixed with 2% of the corresponding mCherry tagged protein at a concentration of 25 µM, phase separation was induced in a test tube for 30 min in the presence of 6% PEG in the well. Experiments were performed on a Zeiss Axio Observer inverted microscope equipped with a Yokogawa CSU-X1-A1 Nipkow spinning disc unit (Visitron Systems; pinhole diameter 50 µm, spacing 253 µm), sCMOS camera (Pco.edge 4.2) and a Plan-Apochromat 63x/1.4 Oil DIC objective. Images were conducted every 5 s for a time course of 10 min with 80% HX, 50 ms exposure and 10% 561 nm laser intensity exposed for 100 ms. Per condition, 3 condensates were bleached after 2 frames with 100% 561 nm laser power for 10 ms per pixel. Image processing was performed in ImageJ selecting the FRAP ROI and two ROIs of the same size within a non FRAPed condensate for bleaching correction and an area without condensate for background correction for every FRAPed condensate. The background value was subtracted from the FRAP and bleaching value, followed by calculating the bleaching factor to correct the FRAP values leading to the final normalization. The normalized FRAP values were fitted to the One-phase association in PRISM.

### Recruitment experiments

Phase separation was induced as described in the "In solution phase separation" section. After 30 min of incubation within the well, the respective mCherry labeled protein was added (at 2% of a total protein concentration of 25 µM) and imaged under the same conditions (on a Zeiss Axio Observer inverted microscope equipped with a Yokogawa CSU-X1-A1 Nipkow spinning disc unit (Visitron Systems; pinhole diameter 50 µm, spacing 253 µm), sCMOS camera (Pco.edge 4.2) and a Plan-Apochromat 63x/1.4 Oil DIC objective) every 5 s for 25 min with 80% HX, 50 ms and 10% 561 laser intensity exposed for 100 ms. Image processing was performed in ImageJ.

## Modelling of disordered regions

The protein structure of the human IRE1α core Lumenal Domain (cLD) dimer was obtained from the Protein Data Bank (www.rcsb.org (Berman et al, 2000), PDB ID: 2HZ6) (Zhou et al, 2006). We added the missing residues (66–70, 89–90, 11–115, 131–152, 308–357, 369–443) as unfolded loops using UCSF Chimera (version 1.15, (Pettersen et al, 2004).

From this model, the regions DR1, DR2 and linker were extracted as isolated peptides and individually mapped to coarse-grained representation.

The peptides obtained were:

**DR1**, 131 – LTGEKQQTLSSAFADSLSPSTS -152;

**DR2**, aa 307-VPRGSTLPLLEGPQTDGVTIGDKGESVITPSTD VKFDPGLKSKNKLNYLRNY- 358;

**Linker region**, aa 369 -LSASTKMLERFPNNLPKHRENVIPAD-SEKKSFEEVINLVDQTSENAPTTVSRDVEEKPAHAPAR-PEAPVDSMLKD - 443.

The conversion of the all-atom models into Martini 3 (Souza et al, 2021) coarse-grained models and the setup of the simulation systems were performed using the tools *martinize2* (https://github.com/marrink-lab/vermouth-martinize) and *insane*.py (Wassenaar et al, 2015) and gromacs/2020.5 tools (*gmx insert-molecules*). The termini were neutralized and the side chain fix was applied to prevent unrealistic side chain orientations as proposed in (Herzog et al, 2016).

## Molecular dynamics simulations

We set up systems containing two disordered regions' peptides by randomly inserting 16 copies of each region in a $30 \times 30 \times 30$ nm³ simulation box. We obtained a system containing DR1 and DR2 and a system containing DR2 and the linker region. We solvated the systems with Martini water molecules and chloride and sodium ions, corresponding to a salt concentration of 150 mM.

After a first energy minimization we equilibrated the system. First, we ran a 10 ps-long simulation using a 1 fs time step and restraining the position of protein backbone beads by using harmonic potentials with force-constants of 1000 kJ mol⁻¹ nm⁻². Afterwards, we ran another 2.1 ns without restraints using a 30 fs time step and a final equilibration of 21 ns. After the equilibration, we ran MD simulations using 20 fs time step. The temperature in the simulation box was controlled by a velocity rescale thermostat (Bussi et al, 2007) (reference temperature $T\_ref = 300$ K, coupling time constant $tau\_T = 1$ ps). The Parrinello-Rahman barostat (Parrinello and Rahman, 1981) (reference pressure $p\_ref = 1$ bar; coupling time constant $\tau\_p = 24$ ps) was used for the last equilibration step and for the production run.

Coarse-grained molecular dynamics simulations were performed using with the Martini 3.0 forcefield (Souza et al, 2021) and the GROMACS 2020.5 software (Lindahl et al, 2019).

## Contact maps from MD simulations

We set up individual simulations for each region (DR1, DR2 and linker) in two different settings, namely containing two peptides or 33 peptides. For the two peptides' simulations, we determined the dimensions of the box by setting a 2 nm distance between periodic images in a cubic box. In the latter simulation setting we randomly inserted 33 peptide copies in a $30 \times 30 \times 30$ nm³ simulation box to

obtain a protein concentration of 2 mM. We solvated the systems with Martini water molecules and chloride and sodium ions, corresponding to a salt concentration of 150 mM.

We analyzed the contacts formed over time among the peptide chains in these simulations. Initially, we computed the contact map between all beads of all peptides at each frame thanks to the Python package Contact Map Explorer (https://github.com/dwhswenson/contact_map, version 0.7.0). Two Martini beads were considered in contact if nearer than 0.5 nm. In the simulations containing 33 copies, we considered which peptide chains are interacting to create a network representation of the clusters at each frame from which we could determine which is the central chain of the cluster. Then we counted all the contacts between beads of the central chain and beads of its neighboring chains at each frame and we averaged over the number of chains interacting with the central one at each frame.

We obtained a matrix of dimensions (*Number of beads per chain*, *Number of beads per chain*) and we convert it to dimensions (*Number of residues per chain*, *Number of residues per chain*) by retaining the maximum score present between all the beads of a pair of residues. The contact matrices were computed in a similar way for systems of two peptides. In these simulations we considered all the interactions happening between the two chains, removing the notion of a central chain. We produced 1D-projections of the contact maps by summing up all the contribution for a specific residue in the final contact matrices for simulations of two or 33 copies.

## Generation of constructs for stable cell lines via lentiviral transduction

For the establishment of stable cell lines in Mouse Embryonic Fibroblasts, a vector with a Tet-On doxycycline-inducible TRE3G promoter was utilized. TRE3G-P2A-eBFP2-PGK-puroSTOP-IRES-rtTA3 (kind gift from Gijs Versteeg) was cut using restriction enzymes BsrGI-HF and BamHI (New England Biolabs). hIRE1α signal sequence with the transmembrane domain (amino acids 1–469) 3XFlag and 6XHis tag and hIRE1α kinase-RNase domain (amino acids 470–977) were amplified frompShuttle-CMV-TO_h-sIRE1-3F6H-GFP-LKR-K36.3. Additionally, due to its higher stability, the GFP tag, upstream of the kinase domain, was replaced with mNeonGreen through Gibson Assembly. The mutations were introduced into IRE1α N-terminal part (amino acids 1–469) or into kinase-RNase domain (amino acids 470–977) via PCR and the mutated fragment were used for Gibson assembly as described above. For the [359]WLLI[362] and D123P construct, overlap extension PCR and site-directed mutagenesis of the wild type lumenal domain plasmid TRE3G-hsIRE1-3F6H-mNeonGreen-LKR-K36.3-PGK was used. All the constructs, except for the core LD, encode for the full-length lumenal sequence of hIRE1α (amino acids 1–469). The lumenal boundaries for the cLD include the core sequence (amino acids 24–389), as well as, a short region proximal to the transmembrane domain (amino acids 434–443), which was shown to be essential for the interaction with the Sec61 translocon (Plumb et al, 2015).

## Transfection of packaging cells

All transfections were performed by mixing DNA and Polyethylenimine (PEI, Polysciences, 23966) in a 1:3 ratio (μg DNA/μg PEI)

in DMEM without supplements. Plasmids for the transfection were purified using an endotoxin-free Plasmid Kit (Qiagen). Transfection was performed using 1100 ng of total DNA (500 ng transfer plasmid, 500 ng pCMVR8.74 Addgene plasmid # 22036, 100 ng pCMV-VSV-G Addgene plasmid # 8454) The day before transfection, $2*10^5$ HEK293T HiEx packaging cells were seeded in 6-well plates in fully supplemented media. The following day, the above-described transfection mixture was added dropwise to the cells. Subsequently, cells were incubated for 48 h.

## Transduction of mouse embryonic fibroblasts (MEF) (IRE1α$^{-/-}$/IRE1β$^{-/-}$) and cell selection

Following the 48 h incubation period, the viral supernatant was sterile filtered with a syringe. The day before transduction $1*10^5$ MEF (IRE1α$^{-/-}$/IRE1β$^{-/-}$) were seeded in a six-well plates in fully supplemented media. For the transduction, the virus was mixed with fully supplemented DMEM (Sigma-Aldrich, D6429) and 8 μg/ml Polybrene (Sigma-Aldrich, TR-1003-G) at 1:50 (v/v). After a 48 h incubation period, cell lines were expanded to 10–15 cm dishes. Protein expression, for subsequent Fluorescence Activated Cell Sorting (FACS), was induced with 400 nM of Doxycycline for 24 h. Cells were sorted in yield mode using BD FACSAria II or BD FACSMelody, gated for low and high-expression cells. The high-expression cells were resorted in stringent mode, following the same procedure. The second FACS sorted high population of the first FACS sorted high population was used for characterization. All the stable cell lines generated in this work are available upon request. The cell lines were not authenticated and tested for mycoplasma contamination.

## Immunofluorescence

IRE1 double-knockout Mouse Embryonic Fibroblasts (MEF) (IRE1 α$^{-/-}$/IRE1β$^{-/-}$) reconstituted with a doxycycline-inducible hIRE1α-mNG (or mutants) were seeded at a density of 20000 cells in a μ-Slide eight-well dish (ibidi) 1 day before the experiment. IRE1α expression was induced for 24 h by adding 400 nM doxycycline. Cells were stressed with 5 μg/ml Tunicamycin for 4 h. The experiment was stopped by washing with cold PBS and fixation with 4% paraformaldehyde for 7 min. After two more washes with PBS, the cells were incubated for 1 h in blocking buffer (PBS, 10% FBS, 1% Saponin) followed by primary antibody incubation overnight at 4 °C (Calnexin, Abcam ab22595 at a dilution of 1:200). After washing twice with wash buffer (PBS, 10% FBS) the secondary antibody (Alexa Fluor 594 goat anti-rabbit, Invitrogen A11037 at a dilution of 1:1000) was incubated for 1 h at room temperature. After three additional wash steps, the sample was imaged in PBS on a Zeiss LSM 980 inverse point scanning confocal microscope with a Plan-Apochromat 63x/1.4 Oil DIC, WD 0.19 mm objective. The microscope is operated by the Zeiss ZEN 3.3 microscope software. mNG and Atto549 were excited by the 488 nm and 561 nm laser diodes of the microscope, respectively. Image processing was performed in ImageJ.

## Western blotting

Treated MEFs at a confluency of 80% were collected in RIPA buffer. The protein concentration was determined by a bicinchoninic acid assay

using a commercially available kit. In all, 10–15 µg protein of the lysate in sample buffer was loaded after denaturation for 10 min at 95 °C on a 10% sodium dodecyl sulfate gel. The proteins were wet transferred from the gel to a nitrocellulose membrane in transfer buffer (25 mM Tris, 192 mM glycine, 20% (v/v) ethanol, pH 8.3) for 120 min at 110 V. The proteins on the membrane were stained with Ponceau S for 5 min followed by blocking in 5% milk for 1 h at room temperature. The primary antibody was applied in 2.5% milk for 1 h at room temperature or overnight at 4 °C. The membranes were washed five times in TBST for 5 min before the secondary antibody in 2.5% BSA (anti-rabbit IgG (H + L), HRP Conjugate, Promega W401B at a dilution of 1:10,000) was added and incubated for 1 h at room temperature. After five 5 min TBST wash steps, the chemiluminescence substrate for the horseradish peroxidase was applied using a commercially available kit. The membranes were imaged using a ChemiDoc system and analyzed with the Image Lab software of Bio-Rad.

Primary antibodies

| Antibody | Dilution | Catalog number | Company |
| --- | --- | --- | --- |
| GAPDH | 1:10,000 | 10494-1-AP | Proteintech |
| IRE1alpha (14C10) | 1:1000 | 3294 | Cell Signalling |

## XBP1 mRNA splicing assays

### Semi-quantitative PCR analyses

The protocol was adapted from (Karagoz et al, 2017) In brief, MEFs grown in a 12-well plate were treated for 24 h with or without 400 nM dox, DMSO or tunicamycin (5 µg/ml) and collected in 180 µl TriFast (VWR Life Science). In total, 100 µl of water and 60 µl of chloroform was added, mixed and incubated for 10 min at room temperature followed by a 5 min 20,800 × g spin. The transparent phase was transferred to a new tube, mixed with 100ul isopropanol and 0.5 µl glycogen and incubated for 15 min on ice. After a 10 min 20,800 × g spin, the pellet was washed three times with 75% ethanol and resuspended in 16 µl water. Alternatively, for some samples the RNA isolation was performed using the High Performance RNA Bead Isolation kit from the MolecularTools shop (VBCF, Vienna) which is based on guanidine thiocyanate lysis steps adapted from Boom et al (Boom et al, 1990) and magnetic beads (GE Healthcare, CAT. #: 65152105050450) and using the Thermo Scientific™ KingFisher™ Flex Purification System according to the kit's protocol. The total RNA concentration was determined by Nanodrop measurement and normalized throughout the samples. The quality of RNA was verified by a 1% Agarose gel. To generate cDNA, total RNA (a minimum of 175 ng) was reverse transcribed using LunaScript RT (New England Biolabs) followed by dilution of 1:5 or 1:10 depending on the normalized RNA input concentration. 4% cDNA product was used to perform semi-quantitative PCR using 50% Taq MM (New England Biolabs) and 0.5 µM of the forward (GAACCAGGAGTTAAGAA-CACG) and reverse (AGGCAACAGTGTCAGAGTCC) primers. The PCR product was amplified for 28 cycles and analyzed on a GelRad stained 3% agarose gel (50:50 mixture of regular and low-melting point agarose). The gels were imaged using a FastGene FAS_V Geldoc System and analyzed with the Image Lab software of Bio-Rad.

### Real-time quantitative reverse transcription PCR analyses

The qPCRs were conducted on a BioRad CFX384 Touch™ Real-Time PCR-System in 384-well plates in triplicate. Per well there was 2 µl cDNA, 0.8 µl of Forward and Reverse primer mix (10 µM concentration each), 5 µl 2× qPCR Master Mix and 2.2 µl nuclease/RNase free water to a total volume of 10 µl. The 2× qPCR Master Mixes used were either 2X HS qPCR Dye Master Mix provided by the VBC core facilities based on Wang et al (Wang et al, 2004) and Dang and Jayasena (Dang and Jayasena, 1996) with reaction conditions adapted from Pellissier et al (Pellissier et al, 2006) or a 2× Master Mix prepared following the protocol by Gijs Versteeg and Adrian Söderholm. Data was processed using the ΔCq method in R with the tidyqpcr v. 1.0 package (Haynes and Wallace, 2021) (additional packages used in the processing are data.table v. 1.14.4, rmarkdown v. 2.24 (Xie, et al, 2018; Xie et al, 2020, https://github.com/rstudio/rmarkdown), rstatix v. 0.7.2, tidytable v. 0.10.1.9, tidyverse v. 2.0.0. The values are plotted as relative fold change of the target normalized to their respective reference gene expression level. Target primers for murine spliced Xbp1 were 5′-CTGAGTCCGAATCAGGTGCAG-3′ for forward and 5′-GTCCATGGGAAGATGTTCTGG-3′ for reverse, taken from Scortegagna et al (Scortegagna et al, 2014). Target primers for murine C/EBP homologous protein (CHOP) were 5′-GGAGCTGGAAGCCTGGTATG-3′ for forward and 5′-GGATGTGCGTGTGACCTCTG-3′ for reverse (Cao et al, 2019). The reference gene for normalization was murine HPRT (Hypoxanthine guanine Phosphoribosyl-transferase) (forward primer: 5′-GCAGTCCCAGCGTCGTGATTA-3′, reverse primer: 5′-TGATGGCCTCCCATCTCCTTCA-3′) from Manakanatas et al (Manakanatas et al, 2022).

# Data availability

All the raw data in this work has been deposited to Biostudies with the accession number: S-BIAD1023. One can access the data through the link below: https://www.ebi.ac.uk/biostudies/bioimages/studies/S-BIAD1023?key=426953ee-3da4-4cf4-a496-e607ce240aff.

The source data of this paper are collected in the following database record: biostudies:S-SCDT-10_1038-S44318-024-00207-0.

# Peer review information

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

## Acknowledgements

We thank late Thomas Peterbauer at the Max Perutz Labs Biooptics Light Microscopy Facility for his help and support. We are grateful to Kitti Csalyi and Thomas Sauer at Max Perutz Labs Biooptics FACS facility for their help. We thank Grzegorz Scibisz and Sertan Atilla for their support with the expression and purification of mCherry-IRE1α LD-10His. We are grateful to Aleksandra S Anisimova with her help in the generation of stable cell lines and the statistical analyses of the data. We thank Venja Vieweger for her help with the characterization of the WLLI and D123P IRE1 mutants in cells. We are thankful to Monika Kubickova for the help with the AUC experiments. We acknowledge CF BIC of CIISB, Instruct-CZ Centre, supported by MEYS CR (LM2023042)) and European Regional Development Fund-Project, UP CIISB" (No. CZ.02.1.01/ 0.0/0.0/18_046/0015974). We thank the members of the Karagöz lab for the critical reading and editing of the manuscript. We are thankful to our colleagues Diego Acosta-Alvear, Vladislav Belyy, Jirka Peschek, Yasin Dagdas, Javier Martinez, Sascha Martens and Alwin Köhler for their invaluable input on the manuscript. We are grateful to Life Science Editors, especially Katrina Woolcock for her useful edits and comments on the manuscript. We acknowledge funding from Austrian Science Fund (FWF-SFB F79 and FWF-W 1261) to GEK. PK acknowledges the support of the Max Perutz PhD fellowship. GAV is funded by Stand-Alone grants (P30231-B, P30415-B, P36572), Special Research Grant (SFB grant F79), and Doctoral School grant (DK grant W1261) from the Austrian Science Fund (FWF). ES and RC acknowledge support and funding by the Frankfurt Institute of Advanced Studies, the LOEWE Center for Multiscale Modelling in Life Sciences of the state of Hesse, the Collaborative Research Center 1507 "Membrane-associated Protein Assemblies, Machineries, and Supercomplexes" (Project ID 450648163), and the International Max Planck Research School on Cellular Biophysics (to RC), the Center for Scientific Computing of the Goethe University and the Jülich Supercomputing Centre for computational resources and support.

## Author contributions

**Paulina Kettel**: Conceptualization; Data curation; Formal analysis; Investigation; Methodology; Writing—original draft; Writing—review and editing. **Laura Marosits**: Conceptualization; Data curation; Formal analysis; Investigation; Methodology. **Elena Spinetti**: Conceptualization; Data curation; Formal analysis; Investigation; Methodology; Writing—review and editing. **Michael Rechberger**: Conceptualization; Data curation; Formal analysis; Investigation. **Caterina Giannini**: Data curation; Formal analysis; Investigation. **Philipp Radler**: Methodology; Project administration. **Isabell Niedermoser**: Data curation; Formal analysis; Writing—review and editing. **Irmgard Fischer**: Data curation; Investigation. **Gijs A Versteeg**: Resources; Supervision; Writing—review and editing. **Martin Loose**: Resources; Supervision; Funding acquisition; Methodology; Writing—review and editing. **Roberto Covino**: Conceptualization; Supervision; Funding acquisition; Investigation; Methodology; Writing—review and editing. **G Elif Karagöz**: Conceptualization; Supervision; Funding acquisition; Investigation; Writing—original draft; Writing—review and editing.

Source data underlying figure panels in this paper may have individual authorship assigned. Where available, figure panel/source data authorship is listed in the following database record: biostudies:S-SCDT-10_1038-S44318-024-00207-0.

## Disclosure and competing interests statement

The authors declare no competing interests.

# Expanded View Figures

**Figure EV1.  Unfolded polypeptides induce clustering of human IRE1α LD on SLBs.**

(A) TIRF images of FRAP experiments of Atto488 labeled DPPE lipids (top) and mCherry-IRE1α LD-10His (bottom) on SLBs showing the dynamic behavior within the indicated time. Scale bar = 5 µm. (B) FRAP curves of mCherry-IRE1α LD-10His tethered to SLBs by 1% Ni-NTA labeled lipids. The SLBs are incubated 10 min with the indicated concentration of the crowding agent PEG before the images are taken. The mobile fraction and diffusion values are decreasing with increasing PEG concentration. (C) FRAP curves displaying the fluorescent intensity of Atto488 labeled DPPE lipids within SLBs treated with the indicated concentration of the crowding agent PEG over time. (D) TIRF images displaying mCherry-10His control and the membrane (Atto488 DPPE) with and without PEG. Scale bar = 5 µm. (E) FRAP curves displaying the fluorescent intensity of Atto488 labeled DPPE lipids within SLBs treated with the indicated concentration of the crowding agent PEG over time. $n = 3$ independent experiments were performed to obtain the data for the FRAP curves. The error bars represent the standard deviation. (F) FRAP curves displaying the fluorescent intensity of mCherry-10His control on SLBs treated with the indicated concentration of the crowding agent PEG over time. $n = 4$ independent experiments were performed to obtain the data for the FRAP curves. The error bars represent the standard deviation. (G) TIRF images of mCherry-hIRE1α LD-10His tethered to SLBs by 1% Ni-NTA labeled lipids in the absence of PEG, in presence of 11% PEG and where PEG is washed out from the well. Scale bar = 5 µm. (H) Amino acid sequences of model unfolded polypeptides MPZ1N and MPZ1-N-2X and the control non-binding derivate MPZ1N-N-2X-RD. (I) Fluorescence anisotropy experiments monitor the interaction of N-terminal fluorescein labeled MPZ1N-2X and its derivative MPZ1N-2X-RD with IRE1α LD. MPZ1N-2X interacts with IRE1α LD at 2 µM affinity, whereas the MPZ1N-2X-RD is impaired in binding. $n = 2$ independent anisotropy experiments were performed to obtain the data for the curves. The error bars represent the standard deviation. (J) Diagram summarizing mCherry-IRE1α LD-10His clustering on SLBs in the presence of peptides at various PEG concentrations. "X" depicts no cluster and "O" cluster formation. (K) FRAP curves of mCherry-IRE1α LD-10His on SLBs in the absence (black curve) and presence of 10 µM MPZ1N (orange curve), 1 µM MPZ1N-2X (red curve) and 1 µM MPZ1N-2X-RD (blue curve) peptides. Curve marks show the mean value, error bars display the standard deviation and the values are fitted to a one-phase association curve. $N = 3$ independent experiments were performed. The error bars represent the standard deviation. (L) FRAP curves illustrate the mobility of Atto488 labeled DPPE lipids within the SLB belonging to the conditions in (K). The color code corresponds to the one used in (K). $n = 3$ independent experiments were performed to obtain the data for the FRAP curves. The error bars represent the standard deviation.

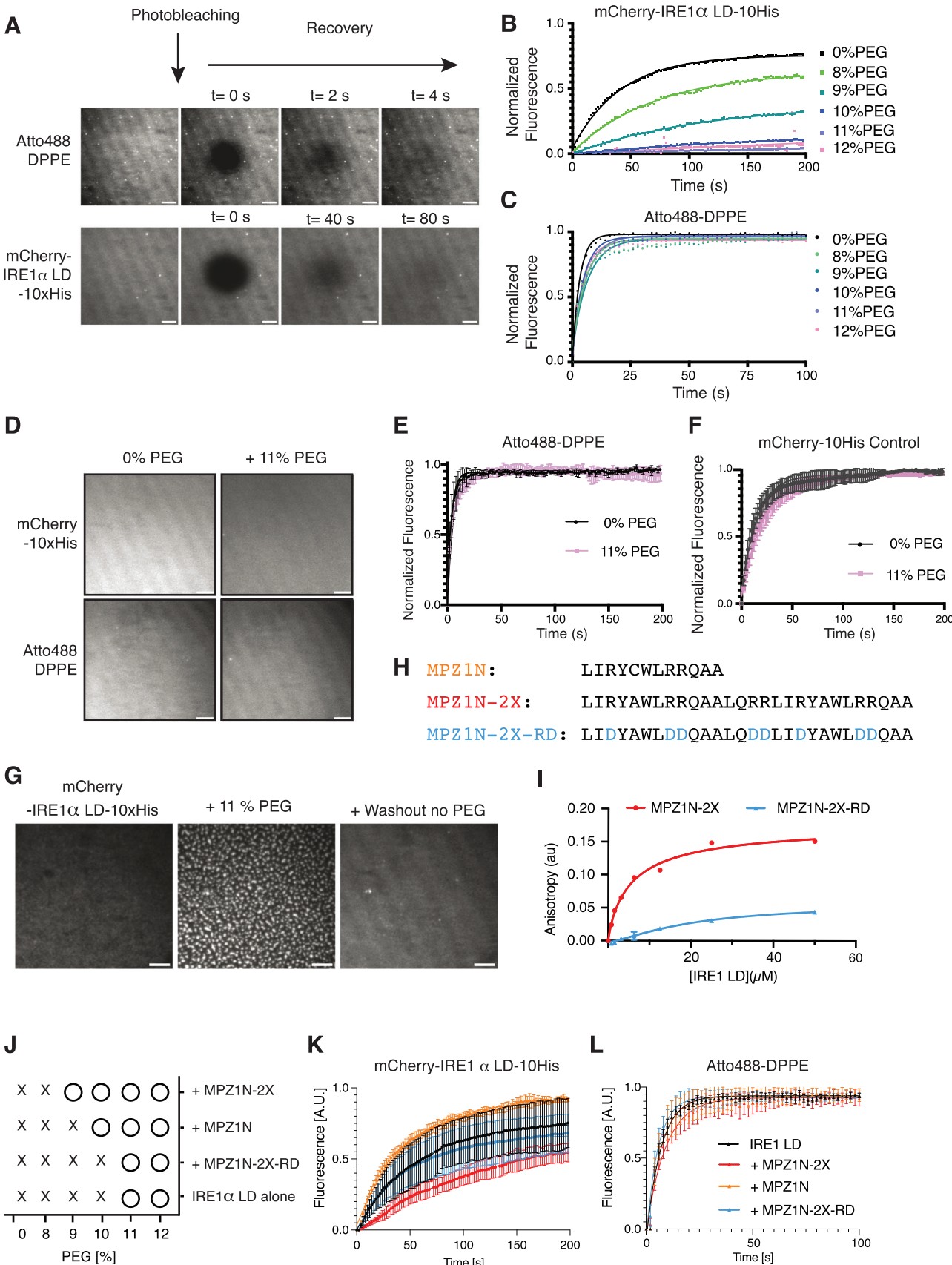

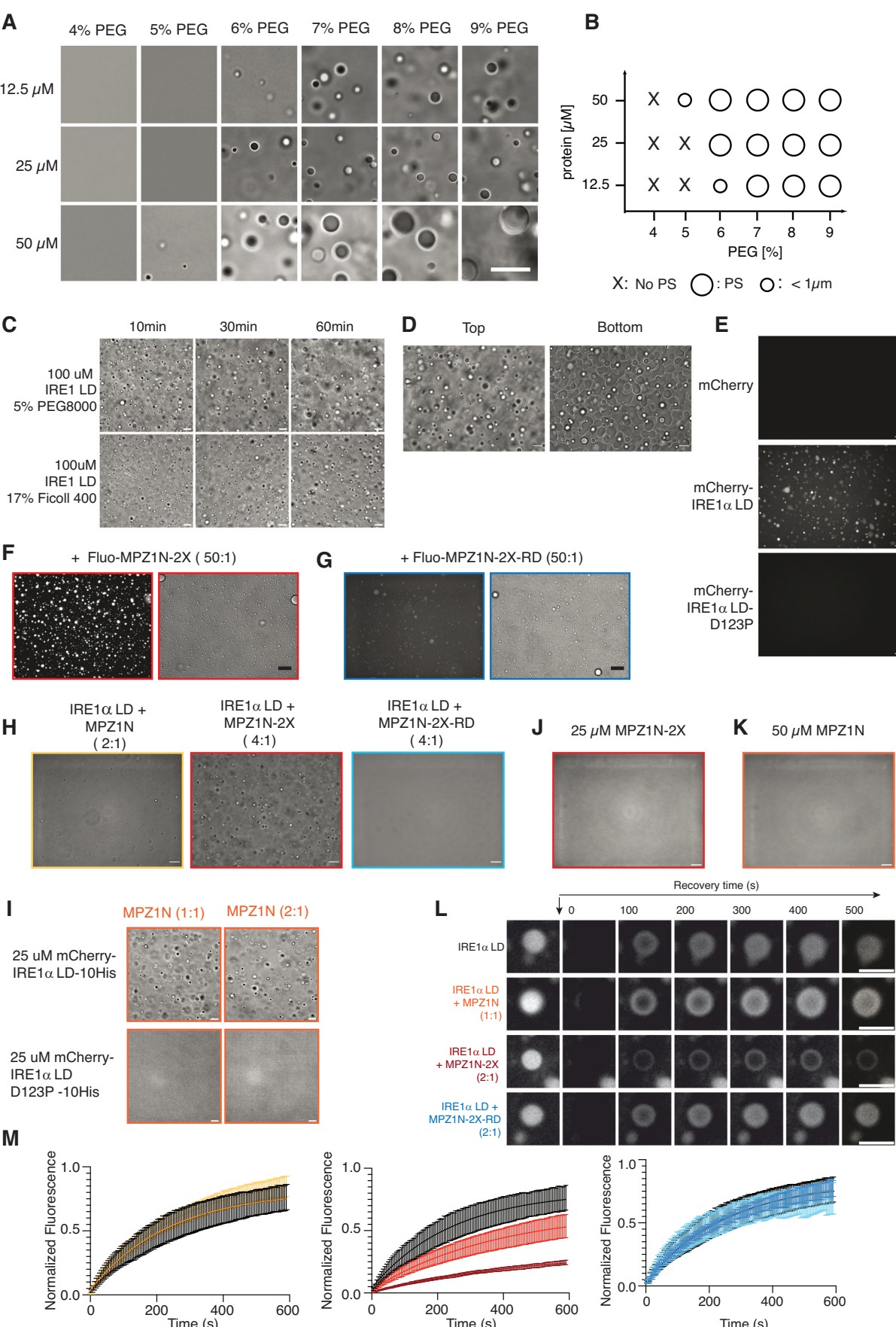

◀ **Figure EV2. Unfolded polypeptide-binding stabilizes human IRE1α LD condensates.**

(A) DIC images of IRE1α LD representing the phase diagram of IRE1α LD. Scale bar = 10 μm. (B) Schematic phase diagram of IRE1α LD condensates at 12.5, 25 and 50 μM at 30 min incubation with 4–9% PEG as in (A). No phase separation (PS) is indicated by a cross and phase separation (PS) is indicated by a circle. The smaller circle refers to condensates with diameter <1 μm. (C) DIC images of 100 μM IRE1α LD in the presence of 5% PEG (top) or 17% Ficoll 400. Scale bar = 10 μm. (D) DIC images of 50 μM IRE1α LD incubated with 6% PEG for 30 min. The images are obtained at the bottom or top of the well. Scale bar = 10 μm. (E) Fluorescence images of 25 μM mCherry-10His control, mCherry-IRE1α LD-10His and the dimerization mutant of IRE1α LD, mCherry-IRE1α LD$^{D123P}$-10His after 30 min incubation with 6% PEG. Scale bar = 10 μm. (F) Confocal (left) and bright field (right) images displaying the recruitment of Fluorescein-labeled MPZ1N-2X (red) peptide into preformed IRE1α LD condensates. Scale bar = 13 μm. (G) Confocal (left) and bright field (right) images displaying the recruitment of Fluorescein-labeled MPZ1N-2X-RD (blue) peptides into preformed IRE1α LD condensates. Scale bar = 13 μm. (H) DIC microscopy images of 50 μM IRE1α LD incubated with MPZ1N (2:1 stoichiometry, left), MPZ1N-2X (4:1 stoichiometry, middle) or MPZ1N-2X-RD (4:1 stoichiometry, right panel) at 30 min after induction of phase separation with 5% PEG. Scale bar = 10 μm. (I) DIC microscopy images of 25 μM mCherry-IRE1α LD (top) or mCherry-IRE1α LD$^{D123P}$ in the presence of 5% PEG and MPZ1N peptide at 1:1 and 1:2 molar ratio. Scale bar = 10 μm. (J) DIC images of 25 μM MPZ1N-2X peptide in the presence of 6% PEG. Scale bar = 10 μm. (K) DIC images of 50 μM MPZ1N peptide in the presence of 6% PEG. Scale bar = 10 μm. (L) FRAP images of a single IRE1α LD condensate in absence and presence of the model unfolded peptides at the indicated stoichiometry taken before and at the indicated time points after photobleaching. Scale bar = 5 μm. (M) FRAP curves of 25 μM IRE1α LD and 6% PEG in the absence (black curve) and in the presence of MPZ1N peptide (2:1 stoichiometry, light orange curve, 1:1 stoichiometry orange curve), MPZ1N-2X peptide (4:1 stoichiometry, red, 2:1 stoichiometry dark red) and MPZ1N-2X-RD control peptide (4:1 stoichiometry, light blue, 2:1 stoichiometry blue). $n = 9$ condensates in 3 independent experiments were performed to obtain the data for the FRAP curves.

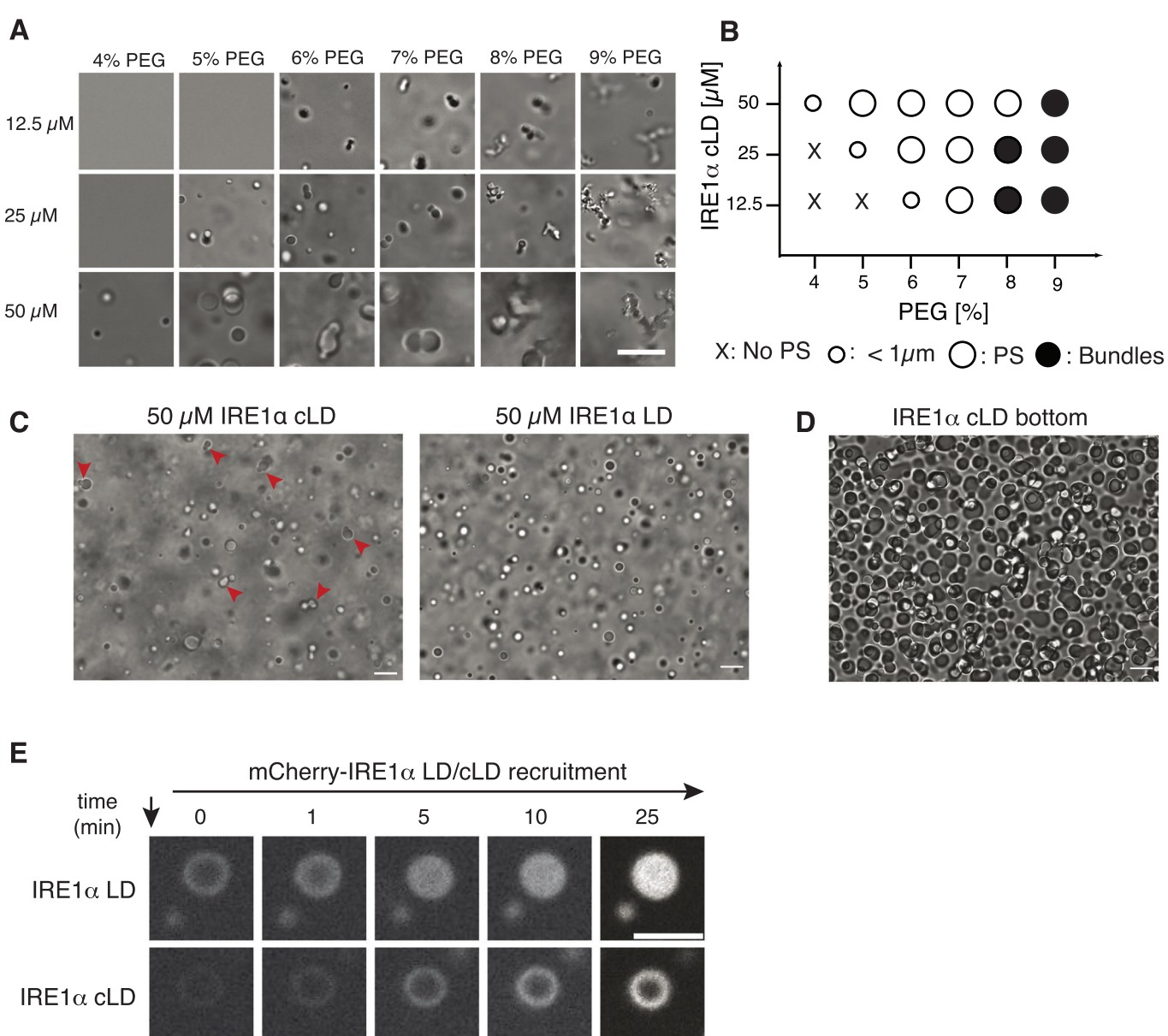

**Figure EV3. Human IRE1α cLD forms rigid condensates.**

(**A**) DIC Images of IRE1α cLD representing the phase diagram at 12.5, 25 and 50 μM acquired after 30 min incubation with PEG at concentrations ranging from 4 – 9%. Scale bar = 10 μm. (**B**) Phase diagram of IRE1α cLD based on images in (**A**). No phase separation (PS) is indicated by a cross, phase separation (PS) is indicated by a circle and condensates that resemble beads on a string are represented by a black circle (bundles). The smaller circle refers to smaller condensates (diameter < 1 μm). (**C**) DIC images of IRE1α cLD (left) and IRE1α LD (right), the condensates that fail to fuse are shown with red arrows. Scale bar = 10 μm. (**D**) DIC images of the bottom of the well of IRE1α cLD (50 μM) condensates taken 60 min after induction of phase separation via addition of 6% PEG showing the phase separation propensity and wetting effect. Scale bar = 10 μm. (**E**) Fluorescence images of 25 μM IRE1α LD (top) or IRE1α cLD (bottom) condensates at the indicated time points after 30 min incubation with 6% PEG following the recruitment of 2% mCherry labeled IRE1α LD or cLD, respectively. mCherry-IRE1α LD-10His is recruited to the center of preformed IRE1α LD condensates, whereas mCherry-IRE1α cLD-10His could only associate with the outer shell of the preformed IRE1α cLD condensates. Scale bar = 5 μm.

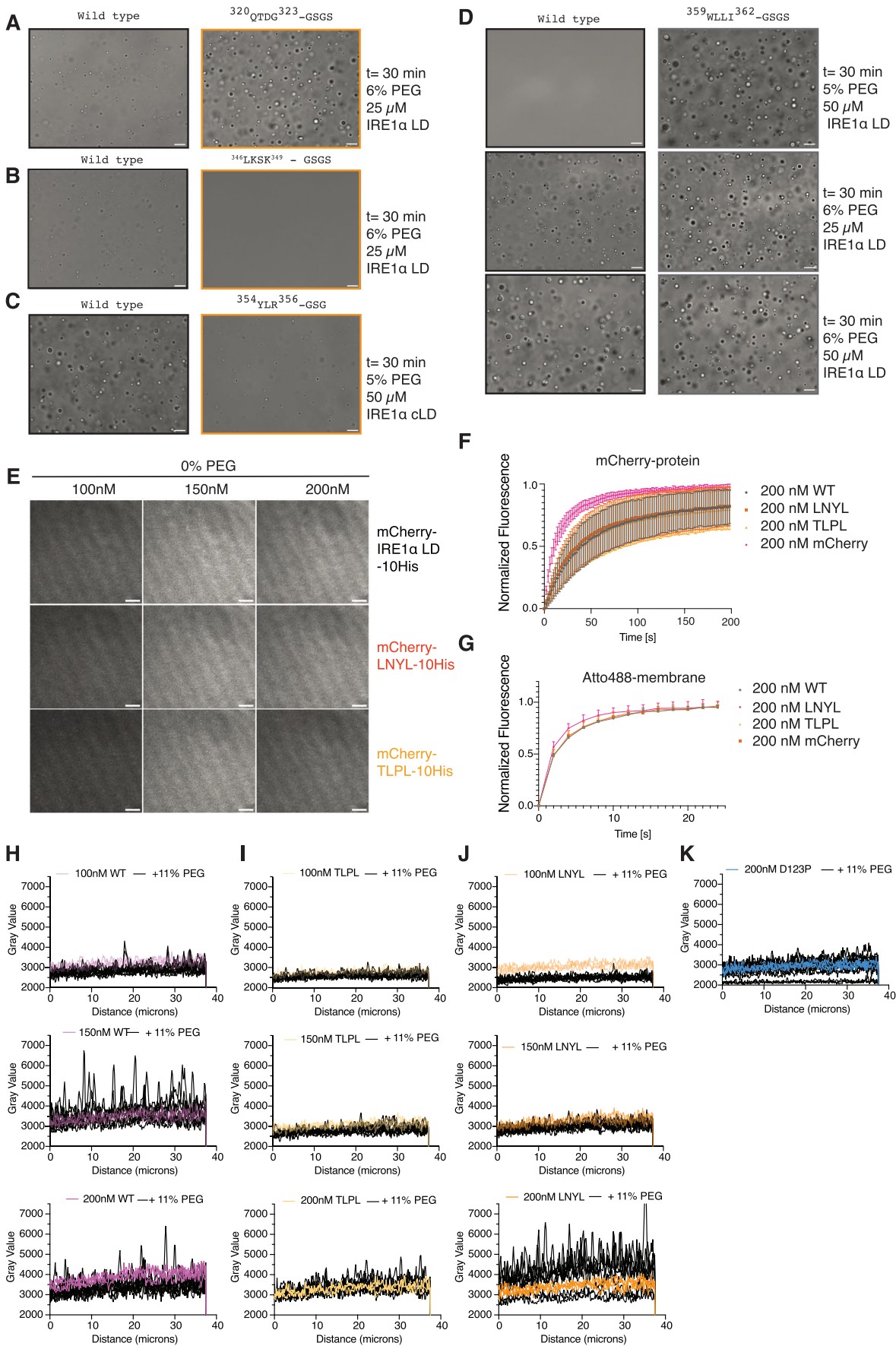

◀  **Figure EV4.  Mutagenesis analyses reveal the critical role of the DRs in IRE1α LD clustering.**

(A) DIC images of 25 μM WT IRE1α LD and IRE1α LD $^{320}$QTDG$^{323}$-GSGS mutant showing LLPS behavior 30 min after induction of phase separation by the addition of 6% PEG. (B) DIC images of 25 μM WT IRE1α cLD and IRE1α cLD $^{346}$LKSK$^{349}$-GSGS mutant showing LLPS behavior 30 min after induction of phase separation by the addition of 6% PEG. (C) DIC images of 50 μM WT IRE1α cLD and IRE1α cLD $^{354}$YLR$^{356}$-GSG mutant showing LLPS behavior 30 min after induction of phase separation by the addition of 5% PEG. (D) DIC images comparing the LLPS behavior of WT IRE1α LD (left column) and IRE1α LD $^{359}$WLLI$^{323}$-GSGS mutant (right column) 30 min after induction of phase separation at 50 μM protein concentration and 5% PEG (top row) at 25 μM protein concentration and 6% PEG (middle row) and at 50 μM protein concentration and 6% PEG (bottom row). Scale bar for all images = 10 μm. (E) TIRF images of mCherry-IRE1α LD-10His (top) and the mCherry tagged mutants $^{352}$LNYL$^{355}$-GSGS (middle) and $^{312}$TLPL$^{315}$-GSGS (bottom) tethered to SLBs by 1% Ni-NTA labeled lipids at concentrations between 100 – 200 nM displaying an evenly distributed fluorescent signal at all concentrations. Scale bar = 5 μm. (F) FRAP curves of mCherry-IRE1α LD-10His, the mCherry tagged mutants $^{352}$LNYL$^{355}$-GSGS and $^{312}$TLPL$^{315}$-GSGS and mCherry control tethered to SLBs by 1% Ni-NTA labeled lipids at a concentration of 200 nM. Curve marks show the mean value, error bars display the standard deviation. $n = 4$ independent experiments were performed. (G) FRAP curves of Atto488 labeled DPPE lipids within SLBs belonging to the tethered proteins in (E). (H) Intensity plot analyses of the mCherry signal in the TIRF images in Fig. 5C for mCherry-IRE1α LD-10His at 100 (top), 150 (middle) and 200 nM (bottom) in the absence (3 lines, colored) and in the presence of 11% PEG (9 lines from three different field of views on the same membrane, black). (I) Intensity plot analyses of the mCherry signal in the TIRF images in Fig. 5C for mCherry-$^{312}$TLPL$^{315}$-GSGS-10His similarly displayed as in (H). (J) Intensity plot analyses of the mCherry signal in the TIRF images in Fig. 5C for mCherry-$^{3352}$LNYL$^{355}$-GSGS-10His similarly displayed as in (H). (K) Intensity plot analyses of the mCherry signal in the TIRF images in Fig. 5D for mCherry-D123P-10His at 200 nM in the absence (3 lines, blue) and in the presence of 11% PEG (9 lines from three different field of views on the same membrane, black).

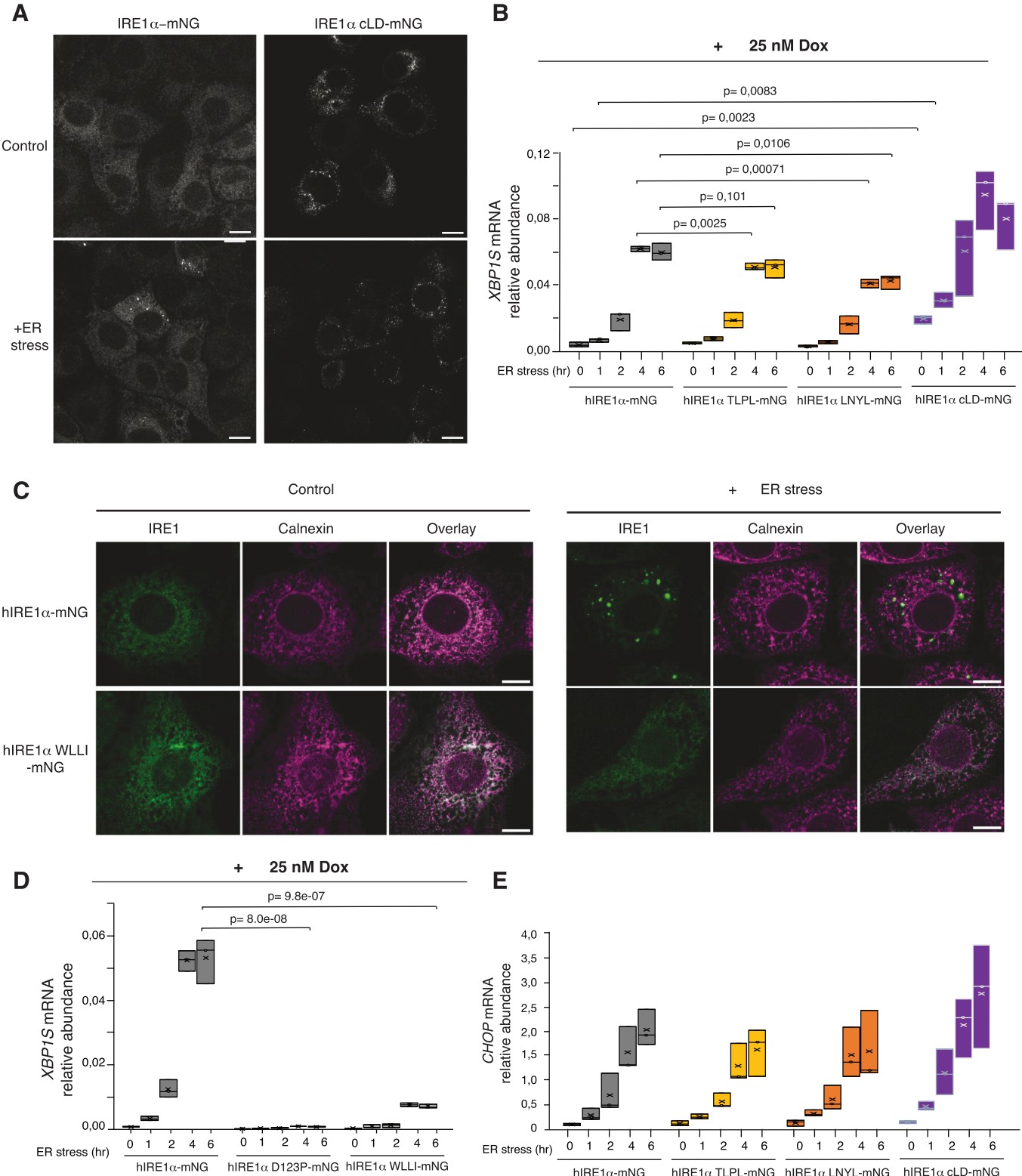

◀  **Figure EV5.  IRE1α LD DR mutants display impaired IRE1α clustering and activity in vivo.**

(A) Immunofluorescence images of MEFs treated with 100 nM doxycycline to induce expression of IRE1α-mNG and the IRE1α cLD-mNG mutant in the absence (top row) of stress and treated with 5 µg/ml ER stressor Tunicamycin for 4 h (bottom row). IRE1α-mNG and its mutants are visualized by mNG fluorescence (green). Scale bar = 10 µm. (B) qRT-PCR to monitor splicing of *XBP1* mRNA by IRE1α-mNG and its mutants at different time points after induction of ER stress by addition of 5 µg/ml Tunicamycin. Cells are treated with 25 nM doxycycline for 24 h to induce expression of IRE1α-mNG and its mutants. Displayed are the values of $n = 3$ independent experiments and the *P* value. The *P* values were determined by a two-sided Student's *t*-test. The data are shown as box plots. The central line in the box plot marks the median and the cross shows the mean, the boxes mark the first and third quartiles and the whiskers display the maximum and the minimum points. (C) Immunofluorescence images of MEFs treated with 400 nM doxycycline expressing IRE1α-mNG or its mutants in the absence (left panel) of stress and treated with 5 µg/ml Tunicamycin for 4 h (right panel). IRE1α-mNG and its mutants are visualized by mNG fluorescence (green) and the ER-chaperone Calnexin is stained by anti-calnexin antibody (purple). Scale bar = 10 µm. (D) qRT-PCR to monitor splicing of *XBP1* mRNA by IRE1α-mNG and its mutants at different time points after induction of ER stress by addition of 5 µg/ml Tunicamycin. Cells are treated with 25 nM doxycycline for 24 h to induce expression of IRE1α-mNG and its mutants. Displayed are the values of $n = 3$ independent experiments and the *P* value. The *P* values were determined by a two-sided Student's *t* test. The data are shown as box plots. The central line in the box plot marks the median and the cross shows the mean, the boxes mark the first and third quartiles and the whiskers display the maximum and the minimum points. (E) qRT-PCR to monitor CHOP mRNA levels in cells expressing IRE1α-mNG and its mutants (400 nM doxycycline) at different time points after induction of ER stress by addition of 5 µg/ml Tunicamycin. Displayed are the values of $n = 4$ independent experiments.

