## [Peer Review File · The EMBO Journal]

Disordered regions in the IRE1 α ER lumenal domain mediate its stress-induced clustering

Paulina Kettel, Laura Marosits, Elena Spinetti, Michael Rechberger, Caterina Giannini, Philipp Radler, Isabell Niedermoser, Irmgard Fischer, Gijs Versteeg, Martin Loose, Roberto Covino, and Gülsün Karagöz

Corresponding author(s): Gülsün Karagöz (guelsuen.karagoez@meduniwien.ac.at)

Review Timeline:

Submission Date:	17th May 23
Editorial Decision:	26th May 23
Appeal Received:	26th May 23
Editorial Decision:	5th Jul 23
Revision Received:	5th Jan 24
Editorial Decision:	22nd Feb 24
Revision Received:	10th Jun 24
Editorial Decision:	25th Jun 24
Revision Received:	2nd Jul 24
Accepted:	23rd Jul 24

Editor: *Cornelius Schneider*

Transaction Report:

Dear Dr. Karagöz,

Thank you for submitting your manuscript "Disordered regions in its ER lumenal domain drives stress-induced clustering of the UPR sensor IRE1 α " to The EMBO Journal. I have now read your study carefully and have discussed the manuscript with other members of the editorial team. I regret to inform you that we have decided not to pursue the publication at The EMBO Journal, but I would like to suggest a transfer to our partner journal Life Science Alliance.

We appreciate the detailed and thorough investigation of the molecular requirements for the higher order assembly of IRE1a in the ER membrane. The manuscript builds on a rich literature of detailed in vitro and vivo studies which have already determined many molecular determinants and details for dimerization and oligomerization of IRE1a also regarding its LD. While we agree that this manuscript extends the previous observations and shows that there are additional novel regions in the LD of IRE1a which are important for IRE1a clustering and phase separation, and which contribute to the unfolded protein response we think that the conceptual advance provided is not sufficient to justify publication at the EMBO Journal.

That being said, we found that your work would be an excellent candidate for our partner journal Life Science Alliance (<http://www.life-science-alliance.org/>; our broad scope Open Access journal published in partnership between the EMBO, Rockefeller University, and Cold Spring Harbor Laboratory Presses). The editors of Life Science Alliance would be pleased to send your manuscript for in-depth peer review; no reformatting is required. We very much hope you will be interested in this option: please follow the link below for transfer. Eric Sawey, Executive Editor of Life Science Alliance (e.sawey@life-science-alliance.org) will be happy to answer any questions you may have.

Thank you for giving us the opportunity to consider this manuscript. I regret that I could not offer better news this time, and I hope that you will consider our transfer offer.

Yours sincerely,

Cornelius Schneider

Cornelius Schneider, PhD
Editor
The EMBO Journal
c.schneider@embojournal.org

** As a service to authors, EMBO Press provides authors with the possibility to transfer a manuscript that one journal cannot offer to publish to another EMBO publication or the open access journal Life Science Alliance launched in partnership between EMBO Press, Rockefeller University Press and Cold Spring Harbor Laboratory Press. The full manuscript and if applicable, reviewers' reports, are automatically sent to the receiving journal to allow for fast handling and a prompt decision on your manuscript. For more details of this service, and to transfer your manuscript please click on Link Not Available. **

Dear Dr. Schneider,

Thanks for your assessment. Unfortunately, we can agree to disagree on your points on the existing rich literature on oligomerization of human IRE1. I do not know which piece of literature you are addressing to but till I published my paper in 2017 in eLIFE, it was debated whether human IRE1 LD could form oligomers at all based the human crystal structure. Mine was the first paper that even showed human IRE1 LD can form oligomers and this is the only paper that defined the possible elements on human IRE1 LD oligomerization. Importantly, apart from describing novel elements both positively and negatively regulating human IRE1 oligomerization, our work identified the role of phase separation as a novel physical state in driving the UPR. I believe that even this discovery on it is own is conceptually ground breaking as our data clearly shows a functional link between IRE1 phase separation and its function.

Thanks again for your time, I hope to work with you in the future.

Best wishes,

Elif

Dear Dr. Karagöz,

Thank you for submitting your manuscript for consideration at the EMBO Journal. We have now received comments from three reviewers, which are included below for your information.

As you will see from the reports, all reviewers find the proposed role of clustering and phase separation of IRE1 α in the unfolded protein response of interest. However, they also raise several points that would need to be addressed, including the use of different crowding agents and additional mutants for the in vitro studies of the IRE1 α LD. Importantly, there are also concerns regarding the high overexpression levels in the in cellulo experiments as well as concerns raised by referee #1 that the TLPL mutant shows little to no effect in cellulo.

Based on the interest expressed in the reports, I invite you to address these issues in a revised version of the manuscript. I think it would be helpful to discuss the revision in more detail via email or phone/videoconferencing - please let me know which option you prefer. I should also add that it is The EMBO Journal policy to allow only a single major round of revision and that it is therefore important to resolve the main concerns at this stage.

We generally allow three months as standard revision time, which can be extended to six months in the case of major revisions. As a matter of policy, competing manuscripts published during this period will not negatively impact on our assessment of the conceptual advance presented by your study. However, please contact me as soon as possible upon publication of any related work to discuss the appropriate course of action. Should you foresee a problem in meeting this deadline, please let us know in advance to discuss an extension.

When preparing your letter of response to the referees' comments, please bear in mind that this will form part of the Review Process File and will therefore be available online to the community. For more details on our Transparent Editorial Process, please visit our website: <https://www.embopress.org/page/journal/14602075/authorguide#transparentprocess>. Please also see the attached instructions for further guidelines on preparation of the revised manuscript.

Please feel free to contact me if you have any further questions regarding the revision. Thank you for the opportunity to consider your work for publication. I look forward to discussing your revision.

Yours sincerely,

Cornelius Schneider

Cornelius Schneider, PhD
Editor
The EMBO Journal
c.schneider@embojournal.org

- a point-by-point response to the referees' comments, with a detailed description of the changes made (as a word file).
- a word file of the manuscript text.
- individual production quality figure files (one file per figure)
- a complete author checklist, which you can download from our author guidelines

(<https://www.embopress.org/page/journal/14602075/authorguide>).
- Expanded View files (replacing Supplementary Information)
Please see out instructions to authors
<https://www.embopress.org/page/journal/14602075/authorguide#expandedview>

We realize that it is difficult to revise to a specific deadline. In the interest of protecting the conceptual advance provided by the work, we recommend a revision within 3 months (3rd Oct 2023). Please discuss the revision progress ahead of this time with the editor if you require more time to complete the revisions.

Referee #1:

The manuscript submitted by Kettel et al. describes that DRs in IRE1a luminal domain regulate the clustering of IRE1a in vitro and in cells. To analyze the mechanism of IRE1a clustering on the ER membrane, they reconstituted in vitro system using mCherry-IRE1a LD associated to SLBs that mimics the membrane. In this system, IRE1a LD clustering in vitro was reproduced using the unfolded model peptides in the presence of PEG as a molecular-crowding agent. In solution (in the absence of SLB), IRE1a condensates displayed more dynamic and LLPS-like characters. After mutagenesis screen using this solution system, the authors proposed that DRs of IRE1a LD contributed the IRE1a LD cluster formation both in vitro and in cells. The ideas proposed are very attractive for understanding the activation mechanism of IRE1a and may be applicable to other transmembrane signal molecules. Furthermore, the biochemical properties of IRE1a LD in solution and/or tethered to SLBs are of great interest for this molecule. Most important, however, is whether the condensed form of IRE1a LD is physiologically functional. In this regard, the data in Fig.5 seriously decreased my assessment for this manuscript.

Major concerns

1. Fig.5A: i) The authors should define the cluster. Because it is very hard for the readers to distinguish between clusters and other condensed dots. ii) In vitro, IRE1a cLD formed bigger clusters than IRE1a LD, but in cells, IRE1a cLD formed smaller and fainter clusters than IRE1a LD. These results are not correlated with in vitro results. iii) The authors should use interface 1,2 mutants. The results would be useful for interpreting the condensate state.
2. Figs.5B,C: i) Splicing efficiency of IRE1a TLPL appears to be similar to that of control IRE1a, but they did not form clusters, the authors described. These results suggest that the cluster formation is not correlated with IRE1a activation. ii) Time course in experiments of B and C is different. The authors should present the same time course, and should present P values in 5C.
3. IRE1a and its derivatives were overexpressed approximately 30-fold above endogenous IRE1a level (Supp.Fig.8). Overexpression of IRE1a caused activation of IRE1a as indicated by XBP1 mRNA splicing (Fig.5B). However, cluster formation of IRE1a was not detected (Fig.5A, Supp. Fig8G). If so, this result also suggests that cluster formation does not correlate with IRE1a activation.
4. The formation of IRE1a LD condensates in solution was promoted by the addition of unfolded polypeptide. To confirm whether unfolded polypeptide binds to the specific groove of IRE1a LD dimer, IRE1a LD interface 1 mutant should be used in Fig.3 experiment.
5. Recent paper by Belly et al (Ref.30) described that IRE1a clusters are not LLPS and instead contain a diffusionally constrained core. The authors should discuss the different conclusion.

Minor point:

- 1 Fig.1 legend: There are two different explanations in 1D.

Referee #2:

In their study entitled Disordered regions in its ER luminal domain drives stress-induced clustering of the UPR sensor IRE1 Kettel, Marosits et al. analyze the biochemical and biophysical basis for the well-known IRE1 clustering behavior upon ER stress. This clustering is relevant for IRE1's biological functions, including RNA splicing, and thus of high cell biological interest. The authors use advanced in vitro techniques to show that the IRE1 luminal domain can undergo LLPS, in solution and on membranes, and that this is modulated by peptide binding, Using their in vitro assays, they identify short disordered regions

within IRE1 that are important for its assembly and then test the impact of those in cells.

Together, this study provides significant conceptual advancements for our understanding of how IRE1 clustering and thus ultimately its biological function is regulated. As such, it should be of interest to the readers of the EMBOJ. Before moving forward, several important points need to be addressed:

Major points:

- How was IRE1 tethered to the model membranes? Was its endogenous linker region used? If not, this needs to be done.
- The authors use PEG to induce crowding. This should be complemented by protein-based crowding (e.g. BSA) at physiologically relevant (ER-like) concentrations.
- The IRE1 LD D123P mutant should also be tested when membrane-tethered to strengthen the links between the 3D and 2D experiments.
- Are IRE1 mutants available that are disrupted in peptide binding? If so, they should also be included as a control in Figure 1; in particular, since the effects are small (1-2% PEG differences); also, addition of Bip, if it bound to the peptide, could be tested if it reversed peptide-induced assembly of IRE1.
- The link between simulations and experiments appears unnecessarily weak. Mutants that disrupt IRE1 clustering should also be tested in the simulations, and vice versa; how are the interaction sites from the simulations used for experimental design?
- The findings on two interaction motifs that drive IRE1 clustering, namely LNYL and TLPL is very interesting; mutants of those should also be tested in the 2D membrane model. Directly moving to cells is a large step, as e.g. also binding to chaperones/co-factors that modulate IRE1 assembly (there are many) could be affected

Minor points:

- In the title, it should rather be "drive" not "drives"
- Is apoptosis in the assembly-deficient IRE1-expressing cells upon ER stress influenced? One could expect this.

Referee #3:

In "Disordered regions in its ER luminal domain drives stress-induced clustering of the UPR sensor IRE1 α ", Kettel et al investigate ER-stress sensor IRE1's propensity to oligomerize and phase-separate, and the ensuing effects on cellular function. The authors begin by establishing that the luminal domain of IRE1 phase-separates on SLBs. They show that this capability is additionally sensitive to a model peptide, phase - separating under less stringent conditions. IRE-LD is further shown to phase-separate in solution. The authors construct a panel of IRE1 mutants, showing that qualities of the disordered region are key to establishing dynamic, liquid-like clusters, and further substantiate their findings with MD simulations. Finally the authors use a cell-culture system to establish the importance of oligomerization on IRE1's function as an ER-stress sensor

The paper addresses an interesting and important question, and the findings look to enhance the understanding of the UPR. The experiments and findings are generally reliable, but I have concerns about the nature of PEG-based experiments, and the over-expression system used in in-vivo assays.

- 1) Use of PEG: The authors use PEG as a "crowder" to induce phase separation, and see a PEG-concentration dependent phase - transition. Adding PEG is a commonly used tool to induce phase separation of 3D droplets, but the use of it to induce domains in membranes is not straight-forward.
- I) The authors did not rule out interactions between PEG and IRE1-LD. Is it possible to monitor the partitioning of PEG into the droplets that form both on bilayers and in bulk. If PEG partitions into the droplets this would be in line with PEG-IRE1 interactions and not a crowding role.
- II) Why not vary the concentration of peptide over a larger range, while holding PEG-concentration fixed?
- III) The concentration of IRE1-LD on the SLBs is quite modest (1% Ni-NTA lipid implies maximum 1 mol % IRE LD, but experiments were below saturation). Would a higher concentration of membrane-tethered IRE1-LD membrane phase separate at lower PEG concentrations? Is it possible to see phase separation on the bilayer without PEG, or are there problems with aggregation? If possible, this would greatly simplify this aspect of results.

The in-vivo experiments utilize an IRE1 over-expression system that is troubling. The authors assert that IRE1 clusters are too small to see at normal expression levels, and hence use ~30x over-expression.

- 1) I do not understand what the authors mean by "too small to see." If they are small but well dispersed (as I think might be expected) than they should be easily visible. If they are small and crowded together, then they would be "hard to see" but this also might be better interpreted as there not being any domains at all. This needs to be clarified.
- 2) Do cells splice XBP1 in the absence of stress at these high expression levels?
- 3) What is the normal range of IRE1 expression in this cell line? i.e how close is 30x over-expression to the normal variation between cell types?
- 4) Do stressed cells see XBP1 splicing activity at normal expression levels? If so this would suggest that clustering / phase separation of "large" clusters is not important. How large would clusters have to be to have an effect on splicing activity? Can

you rule out that the mutants TLPL and LNYL are not forming small, diffraction limited clusters?

The mutational analyses of IRE1-LD are done through solution assays. Is it possible to embed the mutants in a lipid bilayer? Is there any suggestion that these mutants would have different consequences tethered to the membrane versus in solution?

Minor

Lines 299-303: This sentence is confusing, and should be edited to be more precise. What are the IRE1-LD associations in condensates? The authors should say in what direction the linker changes propensity to phase-separate (decreases).

Referee #1:

The manuscript submitted by Kettel et al. describes that DRs in IRE1a luminal domain regulate the clustering of IRE1a in vitro and in cells. To analyze the mechanism of IRE1a clustering on the ER membrane, they reconstituted in vitro system using mCherry-IRE1a LD associated to SLBs that mimics the membrane. In this system, IRE1a LD clustering in vitro was reproduced using the unfolded model peptides in the presence of PEG as a molecular-crowding agent. In solution (in the absence of SLB), IRE1a condensates displayed more dynamic and LLPS-like characters. After mutagenesis screen using this solution system, the authors proposed that DRs of IRE1a LD contributed the IRE1a LD cluster formation both in vitro and in cells.

The ideas proposed are very attractive for understanding the activation mechanism of IRE1a and may be applicable to other transmembrane signal molecules. Furthermore, the biochemical properties of IRE1a LD in solution and/or tethered to SLBs are of great interest for this molecule. Most important, however, is whether the condensed form of IRE1a LD is physiologically functional. In this regard, the data in Fig.5 seriously decreased my assessment for this manuscript.

We thank the referee #1 for their thorough assessment and constructive suggestions.

Major concerns

1. Fig.5A: i) The authors should define the cluster. Because it is very hard for the readers to distinguish between clusters and other condensed dots.

Response: We thank the referee #1 for bringing up this important issue that requires further clarification. The reticulated nature of the ER makes it difficult to assign small condensed dots to specific protein clusters. This is evident in the merge images of Calnexin and IRE1 staining in Fig. 6A, where most of the condensed dots are present in both channels. Therefore, in Fig. 6A, to assess the clustering efficiency of IRE1 LD mutants compared to the wild-type, we used formation of large distinct foci as a readout. All the three mutants we characterized here show clear phenotypic differences according to this criterion. We clarified this now in the text, please see the lines 429-430.

ii) In vitro, IRE1a cLD formed bigger clusters than IRE1a LD, but in cells, IRE1a cLD formed smaller and fainter clusters than IRE1a LD. These results are not correlated with in vitro results.

Response: As the referee points out, IRE1a cLD forms condensates at lower protein and PEG concentrations. *In vitro* the condensate size of cLD is highly comparable to LD. We realized that the example Fig. 3F might be misleading and provided additional example in EV.3C. The FRAP data in Fig. 3G showed slow recovery of the fluorescence signal of IRE1a cLD with lower mobile fraction compared to IRE1a LD indicating that IRE1a cLD condensates are highly rigid. The DIC images showed that IRE1a cLD condensates stick to one another revealing that rigid IRE1a cLD condensates are fusion in competent (EV.3C). Based on those *in vitro* results, we speculate that the fusion deficiency of IRE1a cLD mutant leads to formation of smaller condensates which is reflected as formation of smaller foci in cells. We now depicted the impaired fusion of cLD condensates by red arrows in Supp. Fig. EV.3C and clarified this in the text, please see the lines 288-291 and 453-455.

iii) The authors should use interface 1,2 mutants. The results would be useful for interpreting the condensate state.

Response: As described above, due to the reticulated structure of the ER, it is not trivial to interpret the condensate state for clusters that are not microscopically distinct as described earlier. Distinguishing those states requires more sophisticated microscopy methods such as single particle tracking experiments performed by Belyy et al. eLIFE 2022. Therefore, here we used formation of microscopically visible distinct foci as a phenotypic readout to assess the clustering efficiency of the DR mutants in cells.

2. Figs.5B,C: i) Splicing efficiency of IRE1a TLPL appears to be similar to that of control IRE1a, but they did not form clusters, the authors described. These results suggest that the cluster formation is not correlated with IRE1a activation.

Response: We thank the referee for pointing out this important issue that requires further clarification. Recent work revealed that formation of microscopically visible foci is not necessary for IRE1 activity (Belyy et al, eLIFE 2022, Gomez-Puerta, eLIFE 2022). Using CRISPR-cas9-editing or by expressing fluorescent-tagged IRE1 at concentrations close to its endogenous level, the authors revealed that IRE1 does not form microscopically distinct clusters during ER stress when expressed at endogenous levels in U2OS and HeLa cells, respectively (Belyy et al, eLIFE 2022, Gomez-Puerta, eLIFE 2022.) In one of these publications the authors used sophisticated single-particle tracking experiments to monitor IRE1 self-association. Their data showed that IRE1 forms small clusters in a stress-dependent manner that are not trackable by confocal imaging (Belyy et al, eLIFE 2022). Therefore, here we do not argue that formation of large foci is necessary for IRE1 activation. Based on our *in vitro* results in solution and on SLBs (Fig. 4B and Fig. 5A-C), we anticipate that the TLPL mutant can form small clusters that are under the diffraction limit of light in cells. Based on our in vitro mutagenesis screening in Fig. 4, it is clear that several independent interfaces contribute to LLPS of IRE1 LD and those alternative interaction surfaces could override the

requirement of the TLPL region in cells. Importantly, the quantitative real time PCR analyses, we now performed as quadruplicates indicate that TLPL is slightly but consistently impaired in its *XBP1* mRNA splicing activity compared to IRE1-mNG indicating the importance of this interface in assembling IRE1 into signaling competent clusters (Fig. 6C). We now clarified this point in our manuscript, please see the lines 464-471, 530-539).

ii) Time course in experiments of B and C is different. The authors should present the same time course, and should present P values in 5C.

Response: We now repeated these experiments to obtain consistent time course as in Fig. 6B also performed quadruplicates to obtain p values. These experiments are now updated as Fig. 6C.

3. IRE1a and its derivatives were overexpressed approximately 30-fold above endogenous IRE1a level (Supp.Fig.8). Overexpression of IRE1a caused activation of IRE1a as indicated by *XBP1* mRNA splicing (Fig.5B). However, cluster formation of IRE1a was not detected (Fig.5A, Supp. Fig8G). If so, this result also suggests that cluster formation does not correlate with IRE1a activation.

Response: As pointed out by the referee, even though clustering is important for IRE1 activity, the formation of microscopically visible foci is not required for its IRE1 activation. It seems that we did not make this clear enough in the manuscript and thank the referee for pointing that out. It is known that overexpression of IRE1 leads to its activation, which is reflected in our splicing assay in Fig. 6B (Li et al., 2010 PNAS). We clarified this in text now in line 539-540.

4. The formation of IRE1a LD condensates in solution was promoted by the addition of unfolded polypeptide. To confirm whether unfolded polypeptide binds to the specific groove of IRE1a LD dimer, IRE1a LD interface 1 mutant should be used in Fig.3 experiment.

Response: We thank the reviewer for bringing this point up. We now used the IF1 mutant mCherry-IRE1a LD^{D123P} was to confirm the direct binding of the peptides to the groove. These data showed that addition of MPZ1N to mCherry-IRE1a LD^{D123P} did not lead formation of condensates. We now included it in EV.2I. The incubation of mCherry-IRE1a LD^{D123P} with MPZ1N-2X instead led to the aggregation and we did not include these data in the manuscript.

5. Recent paper by Belly et al (Ref.30) described that IRE1a clusters are not LLPS and instead contain a diffusionally constrained core. The authors should discuss the different conclusion.

Response: Similar to what was shown in Belly et al., 2020 PNAS in mammalian cells, IRE1a LD reconstituted on SLBs does not recover after photobleaching indicating the formation of rigid clusters on model membranes. Based on our data, we propose that liquid-like interactions driven by the DRs allow assembly of stable IRE1a clusters on membranes.

Minor point:

1 Fig.1 legend: There are two different explanations in 1D.

Response: We thank the referee for pointing this out, we now corrected this mistake.

Referee #2:

In their study entitled Disordered regions in its ER luminal domain drives stress-induced clustering of the UPR sensor IRE1 α Kettel, Marosits et al. analyze the biochemical and biophysical basis for the well-known IRE1 clustering behavior upon ER stress. This clustering is relevant for IRE1's biological functions, including RNA splicing, and thus of high cell biological interest. The authors use advanced in vitro techniques to show that the IRE1 luminal domain can undergo LLPS, in solution and on membranes, and that this is modulated by peptide binding. Using their in vitro assays, they identify short disordered regions within IRE1 that are important for its assembly and then test the impact of those in cells.

Together, this study provides significant conceptual advancements for our understanding of how IRE1 clustering and thus ultimately its biological function is regulated. As such, it should be of interest to the readers of the EMBOJ. Before moving forward, several important points need to be addressed:

We thank referee #2 for the appreciation of our work and the thoughtful evaluation.

Major points:

- How was IRE1 tethered to the model membranes? Was its endogenous linker region used? If not, this needs to be done.

Response: We thank the referee for bringing this up. We now clarified this point in the manuscript in lines 110-111. The entire IRE1 LD (24-443) containing the endogenous linker was tethered onto the SLBs via a 10xHis tag to Ni-NTA headgroup lipids.

• The authors use PEG to induce crowding. This should be complemented by protein-based crowding (e.g. BSA) at physiologically relevant (ER-like) concentrations.

Response: We agree that excluding specific interactions between the LD and PEG is crucial. We now used another crowding agent (Ficoll 400)(EV. 2C). The concentration of the ER varies between 100-400 mg/ml and we observed that IRE1 LD undergoes phase separation at 17 % Ficoll 400 at 100uM.

• The IRE1 LD D123P mutant should also be tested when membrane-tethered to strengthen the links between the 3D and 2D experiments.

Response: We now tested whether mCherry-IRE1a LD^{D123P}-10His mutant undergoes phase separation under the same experimental conditions as the mCherry-IRE1a LD-10His on SLBs and found that the mCherry-IRE1a LD^{D123P}-10His mutant does not efficiently assemble into clusters (Fig. 5A,B,D, EV. 5G) indicating that its dimerization is essential to drive valency of IRE1a LD on membranes similar to what we observed in solution. We now added these results to the manuscript please see the lines 366-367.

• Are IRE1 mutants available that are disrupted in peptide binding? If so, they should also be included as a control in Figure 1; in particular, since the effects are small (1-2% PEG differences); also, addition of Bip, if it bound to the peptide, could be tested if it reversed peptide-induced assembly of IRE1.

Response: We agree with the referee that including IRE1 LD mutants that impair its peptide binding would have been very powerful. Unfortunately, such mutants are not currently available. To test, whether BiP leads to dissolution of peptide-induced IRE1 LD clusters, we added BiP to preformed peptide-induced IRE1 LD clusters in the presence of crowding agent on SLBs. Yet, we observed that BiP itself underwent phase separation under those experimental conditions. Due to this behavior, we could not make clear conclusions. Therefore, we decided not to include those results in the manuscript. Please see the images below on BiP phase separation (Figure 1).

Figure 1: The ER chaperone *BiP* undergoes LLPS in a crowded environment. A. DIC image of 1uM *BiP* in presence of 10% PEG after 30min. B. 25uM *BiP* in presence of 5% PEG after 30min

• The link between simulations and experiments appears unnecessarily weak. Mutants that disrupt IRE1 clustering should also be tested in the simulations, and vice versa; how are the interaction sites from the simulations used for experimental design?

Response: We thank the referee for the thorough assessment. The MD simulations revealed that not all disordered regions have the same clustering propensity and provided a general framework that interactions driving LLPS might be formed by particular areas of DR2 while they are more spread out in the linker (Appendix 3A-C). Moreover, the simulations indicated the importance of charged (positive) residues, in addition to hydrophobic and aromatic, that are included in the mutational screening 320-QTDG-323, 350-NKLN-353, 354-YLR-356, 373-TKML-376. However, the site that formed the strongest contacts with other segments in the DR2 predicted by MD simulations was 346-LKSK-349. In the revised manuscript, we now generated the LKSK mutant to experimentally test its role in LLPS. The LKSK-GSGS, mutant was largely impaired in its phase separation underlining the predictive power of the MD simulations. We added the respective in solution experiment to Figure 4B. Moreover, we clarified how MD simulations guided the experimental work in lines 314-316 and 324-327.

• The findings on two interaction motifs that drive IRE1 clustering, namely LNYL and TLPL is very interesting; mutants of those should also be tested in the 2D membrane model. Directly moving to cells is a large step, as e.g. also binding to chaperones/co-factors that modulate IRE1 assembly (there are many) could be affected

Response: We thank the referee for the suggestion. We now included experiments where we characterized

clustering of TLPL and LNYL mutants on the SLBs in Fig. 5A-C. We first performed mass photometry experiments to determine the molecular mass of those assemblies on membranes in the absence of molecular crowding (Fig. 5A,B). These data revealed that up until 100 nM, which was the highest concentration with low enough particle density to allow mass photometry measurements, all the proteins formed mainly dimers. To assess clustering efficiency of the mutants upon molecular crowding, we then reconstituted them on SLB at various protein concentrations (100, 150 and 200 nM) (Fig. 5C). Notably, while clustering efficiency of TLPL and LNYL mutants was lower at 150 nM compared to the wild type protein, the mutants formed clusters similar intensity and number to wild type IRE1 LD at 200 nM. These data showed that in a concentration-dependent manner on model membranes, alternative low affinity interactions could compensate for the mutated segments in TLPL and LNYL to assemble IRE1a LD into clusters, please see the lines 353-399 in the text.

Minor points:

- In the title, it should rather be "drive" not "drives"

Response: We thank the referee for pointing that out, we now corrected this mistake.

- Is apoptosis in the assembly-deficient IRE1-expressing cells upon ER stress influenced? One could expect this.

Response: Our data on earlier time points of the stress do not show a difference in apoptotic potential of the mutants during acute ER stress, suggesting that the PERK branch is not hyperactivated in IRE1 mutant cell lines, we included these results in E.V.6I.

Referee #3:

In "Disordered regions in its ER luminal domain drives stress-induced clustering of the UPR sensor IRE1 α ", Kettel et al investigate ER-stress sensor IRE1's propensity to oligomerize and phase-separate, and the ensuing effects on cellular function. The authors begin by establishing that the luminal domain of IRE1 phase-separates on SLBs. They show that this capability is additionally sensitive to a model peptide, phase - separating under less stringent conditions. IRE-LD is further shown to phase-separate in solution. The authors construct a panel of IRE1 mutants, showing that qualities of the disordered region are key to establishing dynamic, liquid-like clusters, and further substantiate their findings with MD simulations. Finally the authors use a cell-culture system to establish the importance of oligomerization on IRE1's function as an ER-stress sensor

The paper addresses an interesting and important question, and the findings look to enhance the understanding of the UPR. The experiments and findings are generally reliable, but I have concerns about the nature of PEG-based experiments, and the over-expression system used in in-vivo assays.

Response: We thank referee #3 for the constructive feedback and assessment of our work.

1) Use of PEG: The authors use PEG as a "crowder" to induce phase separation, and see a PEG-concentration dependent phase - transition. Adding PEG is a commonly used tool to induce phase separation of 3D droplets, but the use of it to induce domains in membranes is not straight-forward.

Response: We agree that using PEG as a 2D protein crowder is fairly new although PEG and other molecular crowders have been used on SLB systems (Zhao et al 2015 *Macromolecules*, Takatori et al 2023 *PNAS*). As IRE1's soluble LD is driving LLPS, tethering it on the SLBs is crucial in order to observe its clustering behavior closer to its physiological context. The addition of PEG onto the SLBs did not impact the membrane integrity (EV 1.C, F), therefore we think using PEG to mimic the crowding in the ER lumen is a powerful strategy.

l) The authors did not rule out interactions between PEG and IRE1-LD. Is it possible to monitor the partitioning of PEG into the droplets that form both on bilayers and in bulk. If PEG partitions into the droplets this would be in line with PEG-IRE1 interactions and not a crowding role.

Response: We thank the referee to point out an additional control. We integrated in the revised version another assay where we used Ficoll 400 as another crowder. Addition of Ficoll 400 led to formation of IRE1 LD condensates in solution supporting LLPS of IRE1 LD driven by molecular crowding (EV. 2C).

II) Why not vary the concentration of peptide over a larger range, while holding PEG-concentration fixed?

Response: We thank the referee for the suggestion. We performed the experiments with excess peptide, yet under those conditions we observed aggregation and impairment of the integrity of the membranes due to fairly hydrophobic nature of those peptides. Therefore, we were limited by the concentrations used in the manuscript, where we stayed close to the KD.

III) The concentration of IRE1-LD on the SLBs is quite modest (1% Ni-NTA lipid implies maximum 1 mol % IRE LD, but experiments were below saturation). Would a higher concentration of membrane-tethered IRE1-LD membrane phase separate at lower PEG concentrations? Is it possible to see phase separation on the bilayer without PEG, or are there problems with aggregation? If possible, this would greatly simplify this aspect of results.

Response: We thank the referee for pointing out another regime to explore the clustering behavior on SLBs. We increased the protein concentration 5 or 10-fold on SLBs with 3% Ni-NTA lipids. In the presence of 3% Ni-NTA at 2 μ M mCherry-IRE1 α LD-10His, we observed a significant decrease in protein mobility. We anticipated that this is due to formation of stable assemblies among IRE1 LD molecules, as the D123P mutant and mCherry control did not display this behavior. Under those conditions, addition of crowding agent led to spinodal decomposition (See below, Figure 2) which is a phenomenon which occurs if the phase transition is not initiated by nucleation (Banjade and Rosen et al. 2014 elife).

Figure 2: Increased mCherry-IRE1 α LD-10His concentration leads to spinodal decomposition on SLBs. (Left) 200 nM mCherry-IRE1 α LD-10His tethered on a SLB containing 1 % Ni NTA lipids in the presence of 11 % PEG resulting in cluster formation. (Right) 2 μ M mCherry-IRE1 α LD-10His tethered on a SLB containing 3 % Ni NTA lipids in the presence of 11 % PEG resulting in spinodal decomposition

The in-vivo experiments utilize an IRE1 over-expression system that is troubling. The authors assert that IRE1 clusters are too small to see at normal expression levels, and hence use \sim 30x over-expression.

1) I do not understand what the authors mean by "too small to see." If they are small but well dispersed (as I think might be expected) than they should be easily visible. If they are small and crowded together, then they would be "hard to see" but this also might be better interpreted as there not being any domains at all. This needs to be clarified.

Response: We are thankful to the referee for pointing this issue out. Recent work revealed that formation of microscopically visible foci is not necessary for IRE1 activity (Belyy et al, eLIFE 2022, Gomez-Puerta, eLIFE 2022). In one of these publications the authors used sophisticated single-particle tracking experiments to monitor IRE1 self-association. Their data showed that IRE1 forms small clusters in a stress-dependent manner that are not trackable by confocal imaging (Belyy et al, eLIFE 2022). The reticulated structure of the ER makes small clusters difficult to visualize by confocal microscopy. Importantly, the published work suggest that IRE1 is found in equilibrium between stable clusters and a highly mobile population that moves in and out of these clusters (Belyy et al., PNAS, 2020). Several biological systems, which undergo LLPS, do not display large foci trackable by confocal microscopy (Kahn et al., Biophys J 2021). Therefore, here we do not propose that IRE1 requires formation large foci in the ER for activity. We now revised the respective part in the manuscript to clarify this issue, please see the lines 427-4301, 539-540.

2) Do cells splice XBP1 in the absence of stress at these high expression levels?

Response: Overexpression of IRE1 leads to its activation (Li et al 2010 PNAS), which is reflected in our splicing assay in **Figure 6C and D**. We now clarified that in the manuscript, please see the lines 439-441.

3) What is the normal range of IRE1 expression in this cell line? i.e how close is 30x over-expression to the normal variation between cell types?

Response:

The range of IRE1 expression in wild type MEF cells can be seen in **EV. 6C-F** quantified by Western blotting. The human protein atlas shows that IRE1 expression in different cell lines range between 6.5 and 91.4 normalized transcript per million, indicating around 14 fold difference in its expression level across tissues (<https://www.proteinatlas.org/ENSG00000178607-ERN1/cell+line#myeloma> Sun Nov 12, 12:24pm). Furthermore, IRE1 expression levels are dysregulated during pathology and shows up to 50 X difference quantified by western blotting analyses in different multiple myeloma cell types in Harnoss et al. 2019, PNAS. When we use 400nM doxycycline to induce IRE1 expression, we exceed the endogenous expression level of IRE1 in most healthy cell types. We and others have observed that the cluster size of IRE1 directly correlates with its expression levels and we choose this concentration to allow the formation of microscopically visible IRE1 foci as a phenotypic readout to

assess the clustering efficiency of the DR mutants by light microscopy in cells and our results revealed a clear phenotypic difference of those mutants in cells. Importantly, even at high IRE1 expression levels, the mutations introduced to disordered regions impact IRE1 clustering and activity. We now assessed XBP1 splicing activity of the mutants at lower protein expression levels induced by 25 nM doxycycline treatment (around 3 fold of endogenous levels). Importantly, those data show that the DR mutants were impaired in IRE1 activity at low protein levels, similar to higher expression levels (EV. 6H).

4) Do stressed cells see XBP1 splicing activity at normal expression levels? If so this would suggest that clustering / phase separation of "large" clusters is not important. How large would clusters have to be to have an effect on splicing activity? Can you rule out that the mutants TLPL and LNYL are not forming small, diffraction limited clusters?

Response: We thank the referee for pointing this out. We partially addressed that comment in point 3 for referee # 1. We now clarified the interpretation of the results in the discussion (line 521-540). IRE1's XBP1 mRNA splicing activity requires formation of higher order oligomers during proteotoxic stress, yet the minimal size of the clusters sufficient to drive IRE1 activation is currently not known. Several systems that undergo LLPS form diffraction limited clusters in cells (Kahn et al., Biophys J 2021). As the referee pointed out, based on our *in vitro* results, we speculate that the TLPL mutant is able to form diffraction limited clusters in cells, therefore it was only slightly impaired in its XBP1 mRNA splicing activity.

5) The mutational analyses of IRE1-LD are done through solution assays. Is it possible to embed the mutants in a lipid bilayer? Is there any suggestion that these mutants would have different consequences tethered to the membrane versus in solution?

Response: We agree that tethering IRE1a LD mutants onto model membranes is very important to assess their clustering behavior closer to the physiological context. We now included experiments where we characterized clustering of TLPL and LNYL mutants on the SLBs in Fig. 5A-C. We first performed mass photometry experiments to determine the molecular mass of those assemblies on membranes in the absence of molecular crowding (Fig. 5A,B). These data revealed that up until 100 nM, which was the highest concentration with low enough particle density to allow mass photometry measurements, all the proteins formed mainly dimers. To assess clustering efficiency of the mutants upon molecular crowding, we then reconstituted them on SLB at various protein concentrations (100, 150 and 200 nM) (Fig. 5C). Notably, while clustering efficiency of TLPL and LNYL mutants was lower at 150 nM compared to the wild type protein, the mutants formed clusters similar intensity and number to wild type IRE1 LD at 200 nM. These data showed concentration-dependent clustering of the mutant on model membranes, indicating that under those conditions, the formation of alternative low affinity interactions could compensate for the mutated segments in TLPL and LNYL to assemble IRE1a LD into clusters, please see the lines 353-399 in the text.

Minor

Lines 299-303: This sentence is confusing, and should be edited to be more precise. What are the IRE1-LD associations in condensates? The authors should say in what direction the linker changes propensity to phase-separate (decreases)

Response: We thank the referee for pointing this out, we agree we did not clearly state this in the manuscript. We now clarified this point in the revised manuscript, please see the lines 300-305.

Dear Dr. Karagöz,

Thank you for submitting your revised manuscript for consideration by the EMBO Journal. It has now been re-evaluated by all three referees whose comments are enclosed. As you will see, two out of three referees are satisfied and in favor of publication, while one of the referee has several additional concerns. In my view these additional concerns are fair and valid. The additional requested experiments seem reasonable and a more nuanced discussion of the reasons for certain discrepancies between in vivo and in vitro results does not affect the broader conclusions and would therefore benefit the manuscript as a whole.

I would therefore ask you to address these concerns in a revised version of the manuscript but we likely will not require an additional round of peer review to not delay the manuscript further if possible. If you have any additional questions, do not hesitate to contact me.

Thank you for the opportunity to consider your work for publication. I look forward to your revision.

Yours sincerely,

Cornelius Schneider, PhD
Editor
The EMBO Journal
c.schneider@embojournal.org

- a point-by-point response to the referees' comments, with a detailed description of the changes made (as a word file).
- a word file of the manuscript text.

- individual production quality figure files (one file per figure)

- a complete author checklist, which you can download from our author guidelines

(<https://www.embopress.org/page/journal/14602075/authorguide>).

- Expanded View files (replacing Supplementary Information)

We realize that it is difficult to revise to a specific deadline. In the interest of protecting the conceptual advance provided by the work, we recommend a revision within 3 months (22nd May 2024). Please discuss the revision progress ahead of this time with the editor if you require more time to complete the revisions.

Referee #1:

Revised manuscript has been improved, but still raises several concerns. The authors' main claim is that the disordered region of IRE1a plays an important role in the clustering of IRE1a under ER stress. However, the data from in vitro and in vivo by various cluster formation assays are not necessarily coincided with their conclusion.

The following list is the major points in this manuscript.

(1) In vitro experiments:

i) LLPS by DIC assay (Fig.4B, EV5): WLLI>LD(WT)>>TLPL=LNYL

ii) Sedimentation V.(Fig.4C): LD(tetramer)>TLPL=LNYL (dimer)

iii) MP on SLB(Fig.5A): LD=TLPL>LNYL>>D123P

iv) TIRF image on SLB(Fig.5CD, EV5): LD>TLPL=LNYL>>D123P

(2) In vivo experiments (about 30 fold over expression compared to WT):

v) Observation of clustering by confocal microscopy (Fig.6A): LD, cLD>>>TLPL, LNYL

vi) XBP1 mRNA splicing(Fig.6BC): LD, cLD>TLPL>>LNYL

For instance, WLLI and D123P mutants are IF2 and IF1 mutants, respectively, which are deficient in oligomer formation. Therefore WLLI would be expected for the defect of LLPS, but the result clearly indicated that WLLI mutant presented high activity for LLPS. On the other hand, TLPL mutant showed similar phenotype to WT(LD) in experiment iii), suggesting that TLPL forms oligomer like LD, but the results from in vivo were quite different. These results would confuse the readers.

- 1) To clarify whether cluster formation is required for IRE1a activation, measure the splicing efficiency of IF1 and IF2 mutants as a negative control, and add those data to Fig.6B and/or C.
- 2) To understand the model easier for the readers, suitable words should be added. For instance, inactive dimer(left), multivalent weak interacted oligomer(center), and stable cluster(right) et.
- 3) Human cLD showed a constitutive actively phenotype. Yeast Ire1 cLD also formed a constitutive cluster (JCB, 2007, doi: 10.1083/jcb.200704166). Discussion of similar phenotype of cLD would be a beneficial to a broad spectrum of EMBO J readers.

Minor:

1) Fig.3G: It is hard to distinguish which is which,

2) Fig.4 legend: Line 702, "mutations (1-7)" should be replaced by "mutations (1-8)".

Line 709, C should be changed to D.

Fig.4B: The result from mutation 7 should be added.

3) Line 801: "stichometry" should be corrected.

Referee #2:

In their revised manuscript and in their reply, the authors have satisfactorily addressed most of my concerns. Only using a protein instead of a further chemical as a crowder would have been nice, and some mutants show only weak effects as the new data show. This should be made clear in the text of the manuscript. Thereafter, I support publication of this nice study.

Referee #3:

I believe my major concerns have been addressed by the revisions. I still consider the presentation of PEG as only a "crowder" is likely an over-simplification, but appreciate the efforts that the authors have made to clearly identify its influence on the results presented.

Point by Point response to Referee's comments:

Referee #1:

Revised manuscript has been improved, but still raises several concerns. The authors' main claim is that the disordered region of IRE1a plays an important role in the clustering of IRE1a under ER stress. However, the data from *in vitro* and *in vivo* by various cluster formation assays are not necessarily coincided with their conclusion.

The following list is the major points in this manuscript.

(1) *In vitro* experiments:

i) LLPS by DIC assay (Fig.4B, EV5): WLLI>LD(WT)>>TLPL=LNYL

ii) Sedimentation V.(Fig.4C): LD(tetramer)>TLPL=LNYL (dimer)

iii) MP on SLB(Fig.5A): LD=TLPL>LNYL>>D123P

iv) TIRF image on SLB(Fig.5CD, EV5): LD>TLPL=LNYL>>D123P

(2) *In vivo* experiments (about 30 fold over expression compared to WT):

v) Observation of clustering by confocal microscopy (Fig.6A): LD, cLD>>>TLPL, LNYL

vi) XBP1 mRNA splicing(Fig.6BC): LD, cLD>TLPL>>LNYL

For instance, WLLI and D123P mutants are IF2 and IF1 mutants, respectively, which are deficient in oligomer formation. Therefore WLLI would be expected for the defect of LLPS, but the result clearly indicated that WLLI mutant presented high activity for LLPS. On the other hand, TLPL mutant showed similar phenotype to WT(LD) in experiment iii), suggesting that TLPL forms oligomer like LD, but the results from *in vivo* were quite different. These results would confuse the readers.

We thank the Referee #1 for their careful assessment. Based on the points raised by the Referee #1, we realized that some of the main conclusions are not clearly described. As Referee #1 indicated, our data show that the WLLI (IF2) mutant is impaired in oligomerization but still undergoes LLPS, suggesting that IRE1 LD uses different interfaces to form oligomers and undergo phase separation. At the first glance, the differences in the results of the *in vitro* experiments could be interpreted as inconsistencies, but the differences are due to various experimental approaches and conditions that facilitate protein clustering to different degrees. If we put those experiments to a scale, *in solution* AUC-SV experiments would represent the least favorable conditions for protein clustering and SLB experiments where we apply molecular crowding would represent the most favorable. Under the most and the least favorable experimental conditions for protein clustering, the TLPL and LNYL mutants were similarly impaired in clustering and showed almost identical phenotypes to each other. However, they showed clear differences in the mass photometry experiments on SLBs and the phase separation assays in solution, where the LNYL mutant showed a greater deficiency in cluster formation compared to the TLPL mutant. The strength of our experimental approach lies in the use of different assays to be able to distinguish the differences between these mutants, which is in perfect alignment with the data from our cellular work. We clarified these points in the revised manuscript, please see lines 375-378, 398-403, 544-553, 564-575.

1) To clarify whether cluster formation is required for IRE1a activation, measure the splicing efficiency of IF1 and IF2 mutants as a negative control, and add those data to Fig.6B and/or C.

We thank the Referee #1 for the suggestion, we now included these data to the manuscript. Please see **Fig. 6D, EV 6B,D** and lines 453-457, 488-493. In line with the published data (Li et al., PNAS 2010, Karagöz et al., eLIFE 2017), close to the endogenous expression levels (+ 25 nM doxycycline), the IF1 (D123P) and IF2 (WLLI-GSGS) mutants show almost no splicing activity (**EV. 6D**). At higher expression levels (+ 400 nM doxycycline), the IF1 mutant was largely impaired in XBP1 splicing and the IF2 (WLLI) mutant displayed comparable activity to the LNYL mutant under those conditions (**Fig. 6C,D**). The activity of the IF2 mutant (WLLI) is more sensitive to the higher expression levels in cells compared to the TLPL and LNYL mutants. These data further support our *in vitro* results showing that IF2 mutant (WLLI) undergoes phase separation and forms condensates in a protein concentration-dependent manner. Altogether, these new results further underline the importance of the disordered segments in assembling IRE1 into clusters competent in signaling.

2) To understand the model easier for the readers, suitable words should be added. For instance, inactive dimer(left), multivalent weak interacted oligomer(center), and stable cluster(right) et.

We thank the Referee #1 for the suggestion. We used biomolecular condensates to define multivalent low-affinity interactions in solution, stable clusters for long-lived IRE1 assemblies on membranes and oligomers to define assemblies interacting via the canonical oligomerization interfaces. We now modified the text for consistency.

3) Human cLD showed a constitutive actively phenotype. Yeast Ire1 cLD also formed a constitutive cluster (JCB, 2007, doi: 10.1083/jcb.200704166). Discussion of similar phenotype of cLD would be a beneficial to a broad spectrum of EMBO J readers.

We are thankful to the Referee #1 for their suggestion, we now addressed this in the discussion and cited the manuscript, please see the lines 599-601.

Minor:

1) Fig.3G: It is hard to distinguish which is which,

We thank the Referee #1 for pointing this out. We now recolored the figure to make it more distinguishable.

2) Fig.4 legend: Line 702, "mutations (1-7)" should be replaced by "mutations (1-8)".

Line 709, C should be changed to D.

Fig.4B: The result from mutation 7 should be added.

We thank the Referee #1, we now made the indicated corrections and implemented their suggestions to the figure and the legends.

3) Line 801: "stichometry" should be corrected.

Thanks a lot for the input, we now corrected the typo.

Referee #2:

In their revised manuscript and in their reply, the authors have satisfactorily addressed most of my concerns. Only using a protein instead of a further chemical as a crowder would have been nice, and some mutants show only weak effects as the new data show. This should be made clear in the text of the manuscript. Thereafter, I support publication of this nice study.

We thank the Referee #2 for supporting the publication of our work. While the mutants are impaired in clustering under most experimental conditions, they show a different degree of deficiency in their clustering depending on the experimental setup. We now clarified these points in the text.

Referee #3:

I believe my major concerns have been addressed by the revisions. I still consider the presentation of PEG as only a "crowder" is likely an over-simplification, but appreciate the efforts that the authors have made to clearly identify its influence on the results presented.

We thank the Referee #3 for supporting the publication of our work.

Dear Dr Karagöz,

Thank you for submitting a revised version of your manuscript. We find that you have addressed all the additional remarks raised by the referee. There remain only a few mainly editorial points that have to be addressed before I can extend formal acceptance of the manuscript:

1. FUNDING INFO: missing in eJP: Doctoral School grant (DK grant W1261) from the Austrian Science Fund (FWF); Frankfurt Institute of Advanced Studies, the LOEWE Center for Multiscale Modelling in Life Sciences of the state of Hesse, the Collaborative Research Center 1507 "Membrane-associated Protein Assemblies, Machineries, and Supercomplexes", and the International Max Planck Research School on Cellular Biophysics; MEYS CR (LM2023042)) and European Regional Development Fund-Project „UP CIISB" (No. CZ.02.1.01/0.0/0.0/18_046/0015974).
2. Please add up to 5 key words
3. REFERENCE FORMAT: numbered, but it should be alphabetical, 1 author + et al. but it should be 10 authors + et al.
4. Please rename the COI section to "DISCLOSURE AND COMPETING INTERESTS STATEMENT"
5. CRedit has replaced the traditional author contributions section because it offers a systematic, machine-readable author contributions format that allows for more effective research assessment. Please remove the Authors Contributions from the manuscript and use the free text boxes beneath each contributing author's name in our online submission system to add specific details on the author's contribution. More information is available in our guide to authors.
6. DATASET EV LEGENDS: Table 5 should be renamed to Dataset EV1 with the appropriate callout and legend removed from ms file and uploaded as a separate sheet in the Excel file
7. APPENDIX 1 FILE WITH ToC: Appendix figures and Tables 1-4 should be compiled in Appendix PDF with the nomenclature Appendix Figure S1-S4 and Appendix Table S1-S4 with the corresponding callouts and ToC with page numbers on the title page; Appendix figure legends should be removed from ms file and placed below the corresponding figures in Appendix PDF. Supplementary Table legends should be removed from ms file and placed above the corresponding table in Appendix PDF.
8. Please provide source data source data and compile the source data check list.
9. Papers published in The EMBO Journal are accompanied online by a 'Synopsis' to enhance discoverability of the manuscript. It consists of A) a short (1-2 sentences) summary of the findings and their significance, B) 3-4 bullet points highlighting key results and C) a synopsis image that is 550x300-600 pixels large (width x height, jpeg or png format). You can either show a model or key data in the synopsis image. Please note that the image size is rather small and that text needs to be readable at the final size. Please send us this information together with the revised manuscript.
10. Re-use of image between figure 2a (5min) and Figure 4b (5 min IRE to LD). Re-use of image between figure 5d (mCherry / 0%PEG) and Figure EV5a (mCherry 200nM). Both need to be clarified in the corresponding figure legends. See attached Figure Check Report
11. Figure Legends (main + EV): "1. Please note that a separate 'Data Information' section is required in the legends of figures EV 4a-d.
12. Please note that the figure titles are not provided for the figures EV 1, 2, 3, 4, 5, 6. This needs to be rectified.
13. Please note that the figure EV 6i does not contain any statistical parameter, kindly rectify the statistical test related information in the figure legend appropriately.
14. Please note that the figure legend 5d is incorrectly labelled as 5c. This needs to be rectified."
15. Please note that the box plot needs to be defined in terms of minima, maxima, centre, bounds of box and whiskers, and percentile in the legend of figure 6c.
16. Please note that the box plots need to be defined in terms of minima, maxima, centre, bounds of box, and percentile in the legends of figures EV 6h-i.
17. Please note that information related to n is missing in the legends of figures EV 1e-f, i, l; EV 2m.
18. Please note that the error bars are not defined in the legends of figures EV 1e-f, i, l; EV 2m."
19. Please note that the scale bar needs to be defined for figures 2b, d; EV 2c, i; EV 3c.
20. Please note that the red circles are not defined in the legend of figure 1d-e. This needs to be rectified."
21. There are 6 EV figures, but should be up to 5 EV figures
22. There should be only current figures uploaded as Figure files, other version should be saved as Related Manuscript files
23. movie files - playing. They should be renamed to Movie EV1-EV3 with the corresponding callouts, and the legends should be removed from ms file, and zipped with each movie file.
24. Correct Section order: It should be: title page with complete author information, abstract, keywords, introduction, results, discussion, materials & methods, data availability section, acknowledgements, disclosure and competing interests statement, references, main figure legends, tables, expanded figure legends.

With best regards,

Cornelius

Cornelius Schneider, PhD
Editor | The EMBO Journal
c.schneider@embojournal.org

We realize that it is difficult to revise to a specific deadline. In the interest of protecting the conceptual advance provided by the work, we recommend a revision within 3 months (23rd Sep 2024). Please discuss the revision progress ahead of this time with the editor if you require more time to complete the revisions.

Thank you for submitting a revised version of your manuscript. We find that you have addressed all the additional remarks raised by the referee. There remain only a few mainly editorial points that have to be addressed before I can extend formal acceptance of the manuscript:

1. FUNDING INFO: missing in eJP: Doctoral School grant (DK grant W1261) from the Austrian Science Fund (FWF); Frankfurt Institute of Advanced Studies, the LOEWE Center for Multiscale Modelling in Life Sciences of the state of Hesse, the Collaborative Research Center 1507 "Membrane-associated Protein Assemblies, Machineries, and Supercomplexes", and the International Max Planck Research School on Cellular Biophysics; MEYS CR (LM2023042) and European Regional Development Fund-Project „UP CIISB" (No. CZ.02.1.01/0.0/0.0/18_046/0015974).

We now added the missing information.

2. Please add up to 5 key words.

We now added the missing information.

3. REFERENCE FORMAT: numbered, but it should be alphabetical, 1 author + et al. but it should be 10 authors + et al.

We now changed the reference format, please let us know if this is correct.

4. Please rename the COI section to "DISCLOSURE AND COMPETING INTERESTS STATEMENT"

5. CRediT has replaced the traditional author contributions section because it offers a systematic, machine-readable author contributions format that allows for more effective research assessment. Please remove the Authors Contributions from the manuscript and use the free text boxes beneath each contributing author's name in our online submission system to add specific details on the author's contribution. More information is available in our guide to authors.

We now edited this section accordingly.

6. DATASET EV LEGENDS: Table 5 should be renamed to Dataset EV1 with the appropriate callout and legend removed from ms file and uploaded as a separate sheet in the Excel file

We now edited the file name accordingly.

7. APPENDIX 1 FILE WITH ToC: Appendix figures and Tables 1-4 should be compiled in Appendix PDF with the nomenclature Appendix Figure S1-S4 and Appendix Table S1-S4 with the corresponding callouts and ToC with page numbers on the title page; Appendix figure legends should be removed from ms file and placed below the corresponding figures in Appendix PDF. Supplementary Table legends should be removed from ms file and placed above the corresponding table in Appendix PDF.

We now compiled the appendix file according to instructions.

8. Please provide source data source data and compile the source data check list.

9. Papers published in The EMBO Journal are accompanied online by a 'Synopsis' to enhance discoverability of the manuscript. It consists of A) a short (1-2 sentences) summary of the findings and their significance, B) 3-4 bullet points highlighting key results and C) a synopsis image that is 550x300-600 pixels large (width x height, jpeg or png format). You can either show a model or key data in the synopsis image. Please note that the image size is rather small and that text needs to be readable at the final size. Please send us this information together with the revised manuscript.

We generated a synopsis according to the instructions.

10. Re-use of image between figure 2a (5min) and Figure 4b (5 min IRE to LD). Re-use of image between figure 5d (mCherry / 0%PEG) and Figure EV5a (mCherry 200nM). Both need to be clarified in the corresponding figure legends. See attached Figure Check Report

We corrected the Figure 2a to prevent the re-use of image as we have many replicates of the same condition.

However, we do not have re-use of an image in or 5d, Fig. EV5a. Moreover, they do not show images of mCherry control but the WT and the mutants. We are confused about this.

11. Figure Legends (main + EV): "1. Please note that a separate 'Data Information' section is required in the legends of figures EV 4a-d.

It is not clear to us what needs to be included as "Data information".

12. Please note that the figure titles are not provided for the figures EV 1, 2, 3, 4, 5, 6. This needs to be rectified.

We now added the missing information.

13. Please note that the figure EV 6i does not contain any statistical parameter, kindly rectify the statistical test related information in the figure legend appropriately.
We now renamed EV.6 to EV.5 and we did not have Fig. 6i in the previous version. Please check the latest version and let us know if everything is ok.
14. Please note that the figure legend 5d is incorrectly labelled as 5c. This needs to be rectified."
We now corrected this error.
15. Please note that the box plot needs to be defined in terms of minima, maxima, centre, bounds of box and whiskers, and percentile in the legend of figure 6c.
We now added the missing information.
16. Please note that the box plots need to be defined in terms of minima, maxima, centre, bounds of box, and percentile in the legends of figures EV 6h-i.
We now added the missing information.
17. Please note that information related to n is missing in the legends of figures EV 1e-f, i, l; EV 2m.
We now added the missing information.
18. Please note that the error bars are not defined in the legends of figures EV 1e-f, i, l; EV 2m."
We now added the missing information.
19. Please note that the scale bar needs to be defined for figures 2b, d; EV 2c, i; EV 3c.
We now added the missing information.
20. Please note that the red circles are not defined in the legend of figure 1d-e. This needs to be rectified."
We now added the missing information.
21. There are 6 EV figures, but should be up to 5 EV figures, need to change the figure names in the text.
To reduce number of the EV Figures, we now fused the previous EV4 with previous EV5. We renamed the previous EV6 as EV5.
22. There should be only current figures uploaded as Figure files, other version should be saved as Related Manuscript files
We edited the uploaded files accordingly.
23. movie files - playing. They should be renamed to Movie EV1-EV3 with the corresponding callouts, and the legends should be removed from ms file, and zipped with each movie file.
We now changed the titles and the legends of the movie files.
24. Correct Section order: It should be: title page with complete author information, abstract, keywords, introduction, results, discussion, materials & methods, data availability section, acknowledgements, disclosure and competing interests statement, references, main figure legends, tables, expanded figure legends.
We now edited the manuscript to have the correct section order.

Dear Dr. Karagöz,

I am pleased to inform you that your manuscript has been accepted for publication in the EMBO Journal.

Yours sincerely,

Cornelius Schneider, PhD
Editor
The EMBO Journal
c.schneider@embojournal.org
